# Challenges to *Cannabis sativa* Production from Pathogens and Microbes—The Role of Molecular Diagnostics and Bioinformatics

**DOI:** 10.3390/ijms25010014

**Published:** 2023-12-19

**Authors:** Zamir K. Punja, Dieter Kahl, Ron Reade, Yu Xiang, Jack Munz, Punya Nachappa

**Affiliations:** 1Department of Biological Sciences, Simon Fraser University, Burnaby, BC V5A 1S6, Canada; 2Agriculture and Agri-Food Canada, Summerland Research and Development Center, Summerland, BC V5A 1S6, Canada; dieter.kahl@agr.gc.ca (D.K.); ron.reade@agr.gc.ca (R.R.); yu.xiang@agr.gc.ca (Y.X.); 33 Rivers Biotech, Coquitlam, BC V5A 1S6, Canada; jack.m@3riversbiotech.com; 4Department of Agricultural Biology, Colorado State University, Fort Collins, CO 80523-1177, USA; punya.nachappa@colostate.edu

**Keywords:** bioinformatics, cannabis, hemp, hop latent viroid, molecular diagnostics, plant pathogens

## Abstract

The increased cultivation of *Cannabis sativa* L. in North America, represented by high Δ^9^-tetrahydrocannabinol-containing (high-THC) cannabis genotypes and low-THC-containing hemp genotypes, has been impacted by an increasing number of plant pathogens. These include fungi which destroy roots, stems, and leaves, in some cases causing a build-up of populations and mycotoxins in the inflorescences that can negatively impact quality. Viroids and viruses have also increased in prevalence and severity and can reduce plant growth and product quality. Rapid diagnosis of the occurrence and spread of these pathogens is critical. Techniques in the area of molecular diagnostics have been applied to study these pathogens in both cannabis and hemp. These include polymerase chain reaction (PCR)-based technologies, including RT-PCR, multiplex RT-PCR, RT-qPCR, and ddPCR, as well as whole-genome sequencing (NGS) and bioinformatics. In this study, examples of how these technologies have enhanced the rapidity and sensitivity of pathogen diagnosis on cannabis and hemp will be illustrated. These molecular tools have also enabled studies on the diversity and origins of specific pathogens, specifically viruses and viroids, and these will be illustrated. Comparative studies on the genomics and metabolomics of healthy and diseased plants are urgently needed to provide insight into their impact on the quality and composition of cannabis and hemp-derived products. Management of these pathogens will require monitoring of their spread and survival using the appropriate technologies to allow accurate detection, followed by appropriate implementation of disease control measures.

## 1. Introduction

Microbial plant pathogens, which include fungi, bacteria, phytoplasmas, viruses, and viroids, affect a broad range of plant species worldwide, causing significant losses in the growth and quality of both food and nonfood plants. Effective management of these pathogens requires a timely diagnosis to confirm their involvement in any disease symptoms, which frequently can be problematic where plants fail to show classic visible symptoms, yet show a decline in growth and productivity over time due to underlying infections. Accurate diagnosis in most cases has relied on the utility of molecular diagnostic approaches, most of which are based on the polymerase chain reaction (PCR).

In 2018, the approval of the Farm Bill in the USA that allowed the production of hemp (containing <0.3% THC), and the legalization of cannabis (marijuana) (representing high-THC genotypes) in Canada for medicinal and recreational purposes, sparked interest in expanding the commercial production of these plants, both designated as *Cannabis sativa* L. and members of the Cannabaceae family. To date, more than 100 pathogens/diseases have been identified to cause problems on these two crops [1,2], the majority of which are fungi and oomycetes, followed by viruses and viroids. In this study, we illustrate how a range of PCR-based diagnostic approaches can be used to identify the pathogens of importance affecting cannabis and hemp in North America. We additionally have applied methods in whole-genome sequencing and bioinformatics to better understand the diversity and impact of specific pathogen groups affecting these plants. The diseases and pathogens which are currently of most significant economic concern for cannabis and hemp producers include fusarium root rot (*Fusarium oxysporum*), pythium root rot (*Pythium myriotylum*), powdery mildew (*Golovinomyces ambrosiae*), botrytis bud rot (*Botrytis cinerea*), hop latent viroid, and beet curly top virus [1]. These six pathogens were included in this study as examples of how molecular diagnostic approaches can provide rapid identification and can be used to understand pathogen dynamics and spread from an epidemiological perspective. In addition, a more detailed investigation on the distribution and levels of hop latent viroid (HLVd) within cannabis plant tissues was conducted using several different molecular methods, and the results are presented.

The results presented demonstrate how whole-genome sequencing (NGS) approaches applied to cannabis and hemp plants have revealed insights into the virome of these two crops and illustrate the complex group of viruses and a viroid that are present. Additional studies are likely to reveal additional pathogens that belong to this group. The complex of fungal/oomycete pathogens recovered from symptomatic plant tissues has been identified using PCR with a universal set of primers, and the methods are described. Additional diagnostic approaches based on RT-PCR, RT-qPCR, and ddPCR, as well as LAMP assays, have shown how these methods can be used to characterize the incidence and severity of *Hop latent viroid* (HLVd) affecting cannabis plants. Lastly, this study illustrates the application of molecular diagnostics and bioinformatics to characterize populations of HLVd and *Beet curly top virus* affecting cannabis and hemp crops.

## 2. Results

### 2.1. Detection of Fungal and Oomycete Pathogens on Cannabis

#### 2.1.1. Symptoms and Pathogen Isolation

The symptoms observed on the cannabis plants that were included for analysis are shown in Figure 1. They included stunting and yellowing on vegetative and flowering plants and internal stem discoloration (Figure 1a–c). In some instances, visible growth of the fungal mycelium could be seen on inflorescence tissues (Figure 1d–f). These symptoms were observed on a range of different cannabis genotypes that were cultivated during experiments conducted in 2020–2022. Following surface sterilization and plating onto potato dextrose agar containing 140 mg/L streptomycin sulfate, emerging colonies were purified through subculturing and examined for spore morphology and other features to identify them to genus and species. The appearance of the colonies of the fungi recovered from symptomatic tissues are shown in Figure 1g–l. They confirm the presence of *Fusarium*, *Pythium*, and *Botrytis* species, which was confirmed by molecular analysis.

#### 2.1.2. Molecular Detection by PCR

The primers UN-UP18S42 and UN-LO28S576B amplified a region of the internal transcribed region (ITS1-5.8S-ITS2) of ribosomal DNA and produced bands of different molecular weight sizes when the DNA extracted from diseased and healthy leaf and inflorescence tissues was used (Figure 2). Sequencing of these bands confirmed the presence of *Golovinomyces ambrosiae* (Figure 2a) and *Botrytis cinerea* (Figure 2b) in the affected tissues, as well as the cannabis DNA sequence. When these primers were used with DNA extracted from pure cultures recovered from symptomatic tissues, a range of band sizes was observed, depending on the culture (Figure 3). Sequencing of these bands and comparison to ITS1-5.8S-ITS2 sequences from the National Center for Biotechnology Information (NCBI) GenBank database using BLAST confirmed the identification of various species using cut-off values >99%. The pathogens identified were *Fusarium oxysporum* (Figure 1g), *F. proliferatum* (Figure 1h), *Pythium myriotylum* (Figure 1i), *G. ambrosiae* (Figure 1j), *B. cinerea* (Figure 1k), and *F. sporotrichiodes* (Figure 1l). Two additional fungi—*Trichoderma asperellum* and *Penicillium olsonii*—recovered from diseased root and stem tissues, respectively, were also identified. 

### 2.2. Detection of Viral Pathogens on Cannabis

#### 2.2.1. Symptoms

Leaves displaying symptoms of mosaic, mottling, and line patterns that superficially resembled those caused by viral infections were collected from greenhouse-grown cannabis plants during 2021–2022.

Representative symptoms on leaf samples on several genotypes are shown in Figure 4. These symptoms were consistently observed on vegetative and flowering plants of four genotypes—‘OG Kush’ (OG) (Figure 4a), ‘Headband’ (HB) (Figure 4b), ‘Golden Papaya’ (GP) (Figure 4c), and ‘Motor Breath’ (MB) (Figure 4d). The leaf tissues from these symptomatic plants were subjected to several diagnostic approaches, as described below, to determine if viral pathogens were present. By comparison, leaves with chlorotic sectors and loss of chlorophyll characteristic of somatic mutations are shown in Figure 4e.

#### 2.2.2. PCR with Broad-Spectrum and Specific Primer Sets

Several broad-spectrum degenerate primer sets were tested following the conditions described in Appendix A. As well, specific primers designed to detect turnip ringspot virus, alfalfa mosaic virus (AMV), yobacco mosaic virus (TMV), and cucumber mosaic virus (CMV) were utilized. Leaves of cannabis genotypes OG and HB, with symptoms as shown in Figure 4a,b, were included in the analysis. None of the primer sets produced a band in the samples originating from cannabis leaf tissues, while the positive controls of the infected tobacco leaves showed the expected band size for these viruses. The RT-PCR results for TMV and AMV are shown in Figure 5a–c. These results indicate these specific viruses were absent in these leaf cannabis tissues.

#### 2.2.3. Transmission Electron Microscopy (TEM)

Under the TEM, rod-shaped particles of TMV were observed in leaf tissues from infected positive-control tobacco plants (Figure 5d), while cannabis leaf tissues showed an absence of any particles resembling viruses in the samples examined.

#### 2.2.4. Host Range Studies

Following inoculation of nine host plant species using ground leaf extracts originating from cannabis genotypes OG and HB with mosaic and line-pattern symptoms, and monitoring plants for symptoms over a period of 3 weeks, no symptoms were observed on any of the host plants tested (Figure 6).

### 2.3. High-Throughput Sequencing (HTS)

The HTS analysis was initially conducted on leaf samples originating from the genotypes OG and HB with symptoms of mosaic and line patterns (Figure 4a,b), and was then extended to an additional six cannabis genotypes (PD, Mac-1, PK, BC, CBD, G54-2). Some of these latter genotypes (PD, Mac-1) showed distinct symptoms of stunted and reduced inflorescence growth (Figure 7a,b). These symptoms were also apparent after the inflorescences were trimmed and dried (Figure 7c,d). A comparison of the fresh weights of the inflorescences from healthy (asymptomatic) and symptomatic plants of five genotypes included in the HTS analysis is shown in Figure 7e. The genotypes that were the most significantly affected were PD, Mac-1, and BC, while PK and CBD did not show a significant reduction in the fresh weight of inflorescence tissues or any obvious symptoms. Following Illumina sequencing and assembly, only two confirmed pathogen sequences were detected in the genotypes HB and OG—hop latent viroid (HLVd) and *Cannabis sativa* mitovirus 1 (CasaMV1). The results are summarized in Table 1. The sequences of HLVd were 100% identical at the nucleotide level to each other and to those in GenBank. The sequences of CasaMV1 were 99.6% identical to the sequences found in GenBank. The sequences from this study have been deposited in GenBank (accession OQ420426 for HLVd from HB and accession OQ420425 for CasaMV1 from HB). When the HTS analysis was conducted on the leaf tissues of six additional genotypes of cannabis (PD, Mac-1, PK, BC, CBD, G54-2), the results were similar, revealing only HLVd and CasaMV1 to be present in all genotypes except for PK.

### 2.4. Molecular Assays for Viroid and Other Viral Pathogens

#### 2.4.1. RNA Extractions, cDNA Preparation, and RT-PCR

##### Hop Latent Viroid

The results from the RNA extractions from the leaf tissues of seven cannabis genotypes, and a check for the RNA quality, showed that all samples yielded satisfactory RNA for conducting RT-PCR (Appendix A). Extracted RNA from leaf tissues was subjected to cDNA preparation and RT-PCR using primers that were developed in this study (see Table 2. Primers for the HLVd amplified bands of sizes 256 bp and 512 bp from the leaf tissues of plants of genotype PD showed symptoms of stunted growth and reduced inflorescence development (Figure 8). Sequencing results confirmed that both bands represented HLVd. Tissues from genotype PK did not show a band and there were no symptoms on this genotype (Figure 7). When the RT-PCR analysis was conducted on an additional six genotypes of cannabis, HLVd was shown to be present in PD, Mac, BC, and HB (Figure 9). Three of these four genotypes previously displayed symptoms of stunting and had reduced inflorescence growth, as shown in Figure 7. These results suggest that HLVd infection had impacted their growth. The corresponding band size (256 bp) in genotypes G54-2 and CBD was very faint (Figure 9). Genotype CBD did not display any symptoms attributed to HLVd, and its growth was not affected, as shown in Figure 7. The growth of genotype G54-2 was not measured.

##### Mitovirus

To assess the extent to which CasaMV1 is present in cannabis tissues, leaves from 34 genotypes originating from 20 indoor-grown plants and 14 from outdoor-grown plants were sampled and used for RNA extraction and cDNA synthesis, as described previously. The forward primer: 5′ CGGTAGGATTGCTCAGTCGG 3′ and reverse primer: 5′ CGAACATGCGGTTCATAGGC 3′ were used to amplify a region of the RNA-dependent RNA polymerase enzyme to produce a 998 bp size product. The PCR conditions were the same as those used for HLVd detection (see Section 4). The results (Figure 10) showed that CasaMV1 was present in 18 out of 20 cannabis genotypes grown indoors (Figure 10a) and in 10 out of 14 genotypes grown outdoors (Figure 10b). Sequence analysis indicated the bands were 100% identical to each other. These sequences showed 99.7% sequence homology to Colorado isolate MT878084 and 99.4% to BK010437.1 from Purple Kush. For an additional four cannabis genotypes, samples of roots, petioles, leaves, and flower tissues were assayed for the presence of CasaMV1, and it was found to be present in all the tissue types analyzed (Figure 10c,d).

#### 2.4.2. Sequence Diversity and Molecular Phylogeny of Hop Latent Viroid

A phylogenetic analysis of HLVd was conducted using seven sequences from hops, two from hemp, as well as four from cannabis (selected from GenBank to represent full-length sequences), and included six from the current study. The analysis showed that no significant sequence diversity was present (Figure 11a). However, one SNP was observed between sequences originating from cannabis genotypes ‘Mac’ and G54-2, where a ‘G’ was ‘A’ (Figure 11b).

#### 2.4.3. Droplet Digital PCR

Quantification of HLVd levels (titer) in the leaves of eight cannabis genotypes was conducted using ddPCR. The genotypes PD, Mac, BC, PK, CBD, G54-2, HB, and OG were the same as those used previously for the RT-PCR (Figure 9). The results are shown in Figure 12. The highest titers (>10,000 genomes per reaction) were seen in PD, Mac, and BC, followed by PDH (healthy) and HB, and a low level (10 genomes) was observed in G54-2. Genotypes PK, CBD, and OG had no detectable levels of viroid. The genotypes PD, Mac, and BC also displayed symptoms of HLVd infection, that included stunting and reduced inflorescence growth, and a reduction in inflorescence fresh weight, as shown previously (Figure 7). By comparison, the genotypes PK and CBD showed no symptoms. In comparing these results to those for the same genotypes obtained by RT-PCR, no band was observed in PK and CBD, and a very faint band was seen in G54-2 (Figure 9). The remaining genotypes (PD, Mac, BC) produced a strong band following RT-PCR. Genotype PDH was confirmed to contain HLVd by both RT-PCR and ddPCR despite being asymptomatic. These results show a positive correlation between the results obtained by RT-PCR and ddPCR for all genotypes tested.

#### 2.4.4. Multiplex Taqman RT-qPCR

Inflorescence tissues from symptomatic plants of genotypes PD and Mac, subjected to the RT-PCR and ddPCR methods above, were also tested using a Taqman RT-qPCR method. In both samples, the C_T_ cycle thresholds were around 18, indicating a high level of viroid titre was present in the inflorescences (Figure 13).

#### 2.4.5. LAMP Assays

The tissues used for this analysis were comprised of 30 leaf, 15 petiole, and 30 root samples taken from 6-week to 10-week-old stock plants representing 11 genotypes of cannabis. These samples were tested by LAMP, and showed that 9 leaf, 5 petiole, and 24 root samples tested positive for HLVd (Table 2). There was no case in which a plant that tested negative in the root tissues was found to be positive in the leaves or petioles. Conversely, many samples that were positive for HLVd in the roots were found to be negative in the leaves or petioles (Table 2). These results show that root samples are the most consistent tissue source for detection of HLVd by LAMP in 6-to-10-week-old plants. All genotypes tested showed the highest frequency of samples to be positive for HLVd from root tissues.

### 2.5. Monitoring Distribution of HLVd in Various Tissues of Cannabis Plants

#### 2.5.1. Distribution within Stock Plants

Leaf samples were obtained from various positions within the canopy of individual stock plants, as well as from roots (Figure 14a,b), and subjected to RT-PCR (Figure 14c,d) as well as ddPCR (Figure 14e). The RT-PCR results showed the presence of multiple bands in most samples tested, varying in size from 256 bp, 512 bp, to 768 bp (Figure 14). Sequencing of these bands confirmed they were all HLVd. In genotype G54-2, the viroid was present in the roots, bottom-canopy leaves, and at the top of the plant, but was barely detectable in the middle-canopy leaves (Figure 14c). In genotype PD, the viroid was present in the roots, bottom-canopy leaves, middle-canopy leaves, and in the top-canopy leaves (Figure 14d). Using ddPCR for the same two genotypes, HLVd was detected in the roots and bottom-canopy leaves of genotype G54-2, but not in the middle- or top-canopy leaves. In genotype PD, HLVd was present in the roots, lower-canopy leaves, and middle- and top-canopy leaves at high concentrations (>10,000 genomes) (Figure 14e).

#### 2.5.2. Distribution of HLVd within Inflorescence Tissues

To quantify the distribution of HLVd within cannabis inflorescences, symptomatic ‘Mac-1’ plants were compared with asymptomatic (healthy) plants (Figure 15a). These plants were selected based on the presence/absence of HLVd as confirmed by RT-PCR (see Figure 9). The terminal inflorescence on symptomatic and asymptomatic plants were each removed and brought back to the laboratory (Figure 15a). Using a scalpel, the fan leaves (FL) and inflorescence leaves (IL) were dissected, leaving the central stripped flower (SF) exposed (Figure 15b). A sample of foliage leaves (PL) taken from the bottom canopy of each plant was included for analysis as well. HLVd presence was confirmed using ddPCR in the small inflorescence leaves (IL), the stripped flower (SF), and in the foliage leaves (PL) of symptomatic ‘Mac’, but not in the fan leaves (FL) (Figure 15d). The highest accumulation of HLVd was seen in the stripped flowers, which are made up of clusters of pistils surrounded by the inflorescence leaves (Figure 15b). In the healthy ‘Mac’ plant, none of the inflorescence samples showed HLVd to be present, but a low level was detected in the foliage leaves by ddPCR (Figure 15d).

#### 2.5.3. Distribution of HLVd within Cuttings from Infected Stock Plants

Vegetative cuttings taken directly from a stock plant of genotype ‘Blue Dream’ confirmed to be infected with HLVd were rooted and tested for HLVd using RT-PCR and RT-qPCR after 14 days (Figure 16a). The results consistently showed that all cuttings were infected with the viroid (Figure 16b), as shown by a band of 256 bp size observed in seven infected cuttings (Figure 16b). This was confirmed by RT-qPCR that showed that four out of seven cuttings tested originating from the infected stock plant contained the viroid at a high titre, with C_T_ values of 15–18 (Figure 16c).

#### 2.5.4. Detection of Pathogens in Fresh and Dried Cannabis Inflorescences

Primers for the ITS1-5.8S-ITS2 region of ribosomal DNA were used to detect fungal pathogens, and primers for HLVd were used on tissue samples obtained from freshly harvested inflorescences and from dried cannabis flowers destined for commercial sale (Figure 17a). Out of 20 samples, 7 were positive for HLVd, 4 samples contained *B. cinerea*, 3 had *F. oxysporum*, and 2 had a powdery mildew (Figure 17b). RT-PCR confirmed the presence of HLVd in 7/10 dried flower samples (Figure 17c).

#### 2.5.5. Detection of HLVd and CasaMV1 in Cannabis Seed

Seeds derived from a cross made between an infected ‘Mac1′ female plant and pollen from a male plant of genotype ‘GPie’ were ground individually, and the extracted RNA was subjected to RT-PCR. The results showed that 14/16 seeds (87.5%) were positive for the presence of HLVd (Figure 18a) and 5/5 seeds tested also contained CasaMV1 (Figure 18b). A sample of 10 seeds from the same batch used in Figure 18 was also subjected to Taqman RT-qPCR using the methods described in Section 4.6. The results are presented in Table 3. They show that viroid levels, as estimated by C_T_ values, differed significantly from one seed and the next, and ranged from 17.1 to 33.66, with a mean of 24.07. The infection rate was 70%, as three out of ten seeds did not contain detectable HLVd (Table 3).

### 2.6. Detection of Beet Curly Top Virus in Cannabis Plants

The symptoms observed on cannabis plants characteristic of infection by BCTV are shown in Figure 19. The uppermost leaves on the plants were curled and twisted and deformed (Figure 19a,b). Similar symptoms were observed on plants in the field (Figure 19c). On some plants, leaf curl symptoms were very severe, causing the plants to appear malformed (Figure 19d–f). RT-PCR with BCTV-specific primers showed the presence of a 1 kb band, confirmed to be the Worland strain in the greenhouse-grown plants (Figure 20).

### 2.7. Diagnostic Approaches to Detect Viral and Viroid Pathogens Affecting Low-THC-Containing Cannabis sativa L. (hemp)

#### 2.7.1. Next-Generation Sequencing (NGS)

Symptomatic field-grown plants, from which leaves were collected for shotgun metagenomic analysis, are shown in Figure 21. These samples were collected from various locations in Colorado during 2019, 2021, and 2022. Among these samples, four viruses and one viroid were identified in 2019 (Table 4). In general, the number of pathogens recovered varied by sampling location, and ranged from two to five. Cannabis sativa mitovirus (CasaMV1) and beet curly top virus (BCTV) were the most commonly found in each year of sampling. HLVd was only present in certain fields sampled in 2019. Citrus yellow-vein-associated virus (CYVaV) and Tobacco streak virus (TSV) were also only detected in 2019 (Table 5). In samples collected in 2021, the viruses that were found Cannabis cryptic virus (CanCV), Alfalfa mosaic virus (AMV), BCTV, and CasaMV1 (Table 4). In 2022, the viruses detected were CasaMV1, BCTV, CanCV, and Tomato bushy stunt virus (Table 4).

#### 2.7.2. RT-PCR with Specific Primer Sets

The majority of the viruses identified through NGS had a high percentage nucleotide identity (>90%; Table 5). Only three of the viruses, CasaMV1, TSV, and CYVaV, were below this threshold. The presence of these low-percentage nucleotide-identity (<90%) viruses was confirmed in the respective samples with RT-PCR analysis (Figure 22). These results confirmed the presence of each of the three viruses in the hemp samples.

#### 2.7.3. Sequence Diversity and Molecular Phylogeny of Beet Curly Top Virus

Phylogenetic analysis based on a fragment of the BCTV coat protein (CP) sequences from the 2019 hemp samples had nucleotide identities to one another between 98.99 and 98.24%. Sequences from this study had a 97–99% sequence identity with cannabis BCTV sequences from Colorado (isolate BCTV-Can; MK803280) and Arizona (isolate BCTV-Can-AZ; MW182244). Phylogenetic analysis revealed that sequences of specific strains of BCTV from hemp formed a distinct cluster that separated BCTV-Wor and BCTV-CO sequences (Figure 23). Strains from other locations in Colorado and elsewhere, available in GenBank, are also shown.

#### 2.7.4. Detection of HLVd in Hemp Seeds and on Thrips

A sample of hemp seeds from a commercial supplier (Figure 24a) was shown to contain HLVd on/in the seed tissues following grinding. The RT-PCR analysis showed the expected band size of 256 bp was present in four out of eight seeds tested (Figure 24b). Another batch of seeds from the same sample was germinated, and the developing seedlings were also determined to be infected with HLVd when both the cotyledons and true leaves were sampled (Figure 24c). In addition, a sample containing eight adult thrips, identified as onion thrips (*Thrips tabaci*) (Figure 24d), observed feeding on the infected plants and causing significant damage (Figure 24e) were also shown to contain the viroid. The viroid was likely located on the mouthparts and on the external surface of the insects (Figure 24b, lane ‘T’). When also assayed by the RT-qPCR method described in this study, the thrips sample was confirmed to be HLVd-positive (C_T_ value of 30.3). Thrips are commonly found within greenhouses in large numbers, and can be detected and quantified using yellow sticky cards placed adjacent to plants (Figure 24f).

## 3. Discussion

Rapid and accurate identification of pathogens affecting cannabis and hemp crops is essential to implement the most appropriate disease-management practices. Conventional diagnostic methods, that include symptomology, microscopic observations of pathogen presence/morphology, culturing techniques for pathogen recovery, and pathogen identification and proof of pathogenicity, have all been successfully and reliably used for the diagnosis of emerging pathogens of cannabis and hemp [1,2]. These methods are now being augmented with molecular advances in nucleic acid-based diagnosis for the characterization of specific pathogens based on their DNA or RNA sequences. These methods primarily involve the polymerase chain reaction (PCR), DNA barcoding, and next-generation and high-throughput sequencing. These molecular approaches are particularly useful in instances where multiple pathogens may be involved in a disease complex, as is commonly encountered in cannabis and hemp crops [1]. They are also useful where disease symptoms may be confused with environmental stresses or damage caused by insect pests. Lastly, molecular diagnostics are important for confirming pathogen presence in instances where plants remain asymptomatic after infection, or if the pathogen cannot be recovered on culture media. This study describes a range of recently developed molecular diagnostic approaches that can be used to confirm the presence of fungal and oomycete pathogens affecting cannabis, as well as to identify a diverse group of viruses and a viroid that affect cannabis and hemp crops during commercial production, all of which are summarized in Section 4.3.1. By sampling and performing the requisite analyses over several years (2020–2023) on samples representing many different genotypes of each crop from different geographical regions, the methods described were validated in several laboratories.

The most prevalent fungal and oomycete pathogens detected on the roots and stems of cannabis plants were *Fusarium* and *Pythium* spp., confirming previous reports of the widespread occurrence of these pathogens in different cannabis growing environments [2,4,5,6,7]. On the leaves and inflorescence tissues, powdery mildew (*Golovinomyces ambrosiae*) and *Botrytis cinerea* were shown to be prevalent pathogens on cannabis, as previously reported [8,9,10,11]. The universal eukaryotic primers also confirmed the presence of *Trichoderma asperellum* originating from root tissues and *Penicillium olsonii* originating from stem tissues. The former is a biological control agent that is frequently applied during greenhouse cannabis production, while the latter is a naturally occurring endophyte found in cannabis stems [2,12]. Cannabis DNA was also amplified with this primer set, which was differentiated by its different molecular weight fragment size and unique sequence that differentiated it from the respective pathogens. The advantage of a universal primer set is that prior knowledge of which pathogens may be present is not required, overcoming a limitation to the use of species-specific primers for undetermined pathogens in a diseased sample. The primer set also confirmed that three of the four most prevalent pathogens could be detected in commercially dried cannabis flower samples following PCR amplification and sequencing. If routine testing of harvested product needs to be implemented, the universal primer set could confirm the presence of these three pathogens, and potentially others that may be present. An important component of cannabis quality assurance is the determination of total yeast and mold (TYM) levels in the final dried cannabis product that reaches the consumer [13,14,15]. While molecular approaches were not investigated or utilized in the present study to determine how TYM could be assessed, previous whole-genome sequencing [16,17,18] and the use of PCR approaches that targeted specific fungal species considered to be of importance have been described [19]. These types of studies should be extended to provide a view of the microbiome within cannabis inflorescences using next-generation sequencing to identify the fungal species potentially posing the most concern for human health [15].

Cannabis leaf samples of many genotypes, which occasionally displayed mosaic, line patterns, and mottling symptoms, particularly on younger leaves, were tested with primers to broad virus groups and to three specific viruses—tobacco mosaic virus (TMV), cucumber mosaic virus (CMV), and alfalfa mosaic virus (AMV)—as described in Appendix A. None of these RT-PCR analyses yielded a PCR product that confirmed the presence of these viruses in over 30 leaf samples displaying these symptoms. Furthermore, a host range study, in which leaf extracts from three symptomatic cannabis genotypes were inoculated onto nine plant species, did not result in local lesion development or other symptoms that would indicate the transmission and presence of putative viruses. Transmission electron microscopy, furthermore, showed no virus particles were present in these tissues. Subsequently, whole-genome sequencing approaches were conducted on eight cannabis genotypes displaying these symptoms and which were sampled at different times. The results showed that the majority of the samples (95–100%) contained only hop latent viroid (HLVd) and a previously reported cannabis mitovirus (CasaMV1) [20]. Neither of these pathogens has been demonstrated to cause foliar symptoms resembling those shown in Figure 4. The characteristic symptoms of HLVd infection are stunting and reduced inflorescence growth [21]. Righetti et al. [22] tested leaf samples from hemp plants displaying mosaic and streak patterns using PCR with specific primers to a large group of viruses, and also performed next-generation sequencing, host range transmission studies, and electron microscopy of symptomatic tissues, similar to what was conducted in the present study. They only reported the presence of Cannabis cryptic virus (CanCV) [22,23], which was present in symptomatic and asymptomatic tissues at varying levels. Their results and ours indicate that these symptoms were not caused by any previously described virus group.

In the present study, CasaMV1 was shown to be present in the leaf, petiole, and root tissues of a majority of cannabis plants (>90%) grown indoors, as well as in most cannabis plants grown outdoors. These samples represented a broad range of genotypes. It was also detected in inflorescence and seed samples derived from an infected mother plant. This confirms the previous finding of CasaMV1 reported to be present in tissues derived from the leaves, inflorescences, seeds, seedlings, and roots of *C. sativa* [20]. There has been no prior research to evaluate the impact of CasaMV1 on cannabis plants, which was also detected in hemp plants in commercial fields [3] and, in the present study, on hemp samples. Mitoviruses are commonly reported to occur in fungi and may cause changes in host physiology, in many cases by altering the mitochondrial structure and function [24]. They have also been shown to be present in a number of plant species, including hemp and hops, without causing any apparent symptoms, and are presumed to be cryptic [20]. In fungi, disruptions in mitochondrial function due to mitoviruses can impact growth and pathogenicity [24]. Alterations in the levels of protein expression have been found in some plants infected by a mitovirus [25]. Further research is needed to determine the significance of the widespread occurrence of CasaMV1 in various tissues of cannabis plants and whether it is indeed cryptic [26] or could be causing mild symptoms.

In cannabis plants grown indoors which are propagated vegetatively for successive generations, the occurrence of coinfections with viruses/viroid presents a challenge in experimentally demonstrating their individual effects. For example, both HLVd and CasaMV1 were found simultaneously in a large number of cannabis genotypes in this study, and were present in both symptomatic as well as in asymptomatic plants. Establishing the roles that each may play in symptom development requires the elimination of one/both of these entities, followed by reinoculation with individual and combined virus/viroid or infectious cDNA clones to observe symptoms and discern if there are any potential interactions between these two coinfecting agents. To date, these types of inoculation experiments have not been performed. Therefore, while the diagnostic assays described in this study provide confirmation of pathogen presence, their causation or involvement in symptom expression remains to be confirmed. In previous experimental inoculations conducted on hemp plants by Keglar and Sparr [27] (summarized by Miotti et al. [28]), it was reported that a number of mechanically transmitted viruses could cause mosaic and mottling symptoms on leaves, including AMV, CMV, potato viruses X and Y, and arabis mosaic virus. None of these viruses were identified in this study on cannabis, while AMV was detected in hemp plants in 2020. In addition, despite widespread conclusions in many nonverifiable sources, such as Internet sites, stating that TMV is an important pathogen causing mosaic symptoms, it has not been demonstrated to infect cannabis or hemp plants to date, and there are no previous reports of its natural occurrence on these hosts.

A possible explanation for the mosaic and mottling symptoms seen on cannabis plants is that they are the result of somatic mutations, which are commonly seen on vegetatively propagated plants [29,30]. A few obvious chimeras were observed on leaves of cannabis plants during this study (Figure 4), some of which bore a striking resemblance to the foliar symptoms putatively attributed to virus infection. Further research is needed to establish whether these symptoms are the result of somatic mutations, which are known to occur in cannabis. Adamek et al. [31] demonstrated the extent of intraplant variation that arises from vegetative propagation from a cannabis stock plant to give rise to genetic mosaicism. Using the deep sequencing of whole genomes, they reported a higher occurrence of variants among shoots obtained from actively growing regions of the plant compared to older shoots at the bottom of the plant. Their results showed that a large number of mutations arise as the cannabis plant grows, and is maintained for a long period of time, and, as a consequence, can potentially impact the functions of important genes [31]. Epigenetic changes in plants and other organisms, caused by DNA methylation and histone modifications, among other mechanisms, have been proposed to be heritable, resulting in these epimutations potentially contributing to phenotypic variation in subsequent generations [32]. In cannabis plants that are extensively propagated through vegetative means, the appearance of these variant phenotypes may be frequent and transgenerational, and may be confused with symptoms of putative infection by viruses such as TMV or CMV.

Distinct symptoms of stunted plant growth, leaf distortion, chlorosis of leaves, and leaf curl have recently been associated with the presence of several viruses confirmed to be present in cannabis and hemp plants. These include Lettuce chlorosis virus (LCV) [33], Beet curly top virus (BCTV) [3,34,35,36,37], and Citrus yellow-vein-associated virus [3,37,38]. Subsequently, diagnostic methods using RT-PCR with specific primers have been developed to detect these viruses [3,33,34,35,36,37,38]. Recent studies have also reported spiroplasma/phytoplasma pathogens affecting hemp, including *Spiroplasma citri* and *Candidatus Phytoplasma trifolii,* which were detected using specific PCR primers [39,40]. Cannabis cryptic virus (CCV) has also been reported in cannabis plants, without causing any apparent symptoms [22,23]. This latter virus was not identified in cannabis samples in this study, but was detected in hemp plants sampled during 2020 and 2021. The potential origins of some of these pathogens can be attributed to infected seed/planting material and/or to influxes of insect vectors. The occurrence of BCTV, HLVd, CasaMV1, Tobacco streak virus (TSV), AMV, Citrus yellow-vein-associated virus (CYVaV), and cannabis cryptic virus (CCV) was confirmed on hemp plants through next-generation sequencing. In addition, Tomato bushy stunt virus was detected for the first time in 2023 and has not been reported previously to infect cannabis or hemp. The diversity of viruses present in commercial hemp crops was much greater compared to indoor-grown cannabis, possibly due to the activity of insect vectors, especially leafhoppers, that are prevalent in outdoor production sites and are known to be vectors of some viruses [3]. BCTV exists as phylogenetically different strains and is reported to affect hemp grown in Colorado, Arizona, Nevada, Oregon, Washington, and California [3,34,35,36,37,39,41]. The virus has an extremely wide host range and is considered to be a pathogen that poses a significant threat to this crop [28]. Additional research is likely to identify more viruses and phytoplasmas occurring on both cannabis and hemp, particularly where these crops are grown in close proximity to nonhemp crops that can provide a source of inoculum for spread to cannabis and hemp plants, in addition to the presence of insect vectors that can transmit these pathogens. A multiplex RT-PCR method that is capable of detecting multiple pathogens simultaneously, including BCTV, HLVd, CYVaV, and CasaMV1, would be useful for rapid diagnostic confirmation and is currently being developed in several laboratories. The various PCR-based diagnostic assays described for these pathogens, augmented by whole-genome and high-throughput sequencing approaches, have been instrumental in confirming the presence and distribution of these pathogens.

Within indoor growing environments used for the majority of commercial production of cannabis, fewer viral pathogens have been reported to date compared to hemp. Mechanical transmission and vegetative propagation are likely the key means through which viral/viroid pathogens would spread indoors if present. Many mechanically transmitted viruses can be transmitted by insect vectors in a nonpersistent manner [42]. Therefore, there is the potential for insect pests reported to occur on indoor-grown cannabis plants, such as rice root aphids (*Rhopalosiphum rufiabdominalis*) and onion thrips (*Thrips tabaci*), to acquire HLVd during feeding. Diagnostic assays utilizing RT-qPCR and RT-PCR have demonstrated that HLVd can be acquired by root aphids (https://www.einnews.com/pr_news/581016978/3-rivers-biotech-identifies-root-aphids-as-potential-vector-for-hop-latent-viroid-hlvd, accessed on 8 October 2023) and potentially by onion thrips, as was demonstrated in this study. A recent study also demonstrated the acquisition of the viroid by leafhoppers feeding on HLVd-infected plants [43], suggesting potential pathogen transmission by these insect species could occur, although the importance of this mode for pathogen transmission remains unknown. In addition, whether HLVd-infected plants serve as better hosts for colonization and development of thrips, as has been demonstrated for Tomato spotted wilt virus-infected tomato plants and western flower thrips [44], remains to be seen.

The presence of whiteflies (*Bemesia tabaci*) in indoor cannabis growing environments was reported to result in transmission of crinviruses, such as Lettuce chlorosis virus (LCV) [33] and Cucurbit chlorotic yellows virus (CCYV) [45], both first detected from cannabis farms in Israel. Transmission of viruses by insect pests affecting hemp crops under field conditions has resulted in multiple occurrences of viral pathogens in one field [37], and sometimes coinfections of up to three different pathogens can occur on a single plant [39]. In Nevada, BCTV was found in association with *Spiroplasma citri* and *Candidatus Phytoplasma trifolii* on the same and on different plants [39]. In Washington, BCTV, HLVd, and CYVaV were reported to occur in the same field and on the same plant [37]. In Colorado and California, multiple viruses and strains have been shown to be present in hemp fields [3,36]. It remains to be seen if multiple viral complexes are detected on indoor-grown cannabis plants. On outdoor-grown cannabis plants, the potential for multiple virus complex development, similar to what has been observed in hemp fields, should be monitored. The development of PCR-based molecular diagnostic assays has aided in the characterization of these pathogen complexes, since symptomology alone is inadequate to distinguish which viruses may be occurring simultaneously.

A large part of this study was focused on developing molecular methods to detect and quantify HLVd in cannabis plants, due to its potential to cause significant damage and for which little is presently understood about its epidemiology and spread [21,46]. Initially, primers used in RT-PCR assays demonstrated the presence of the viroid in stock plants, in rooted cuttings derived from these stock plants, and in various types of plant tissues. The distribution of the viroid was not always uniform in 3–4-month-old infected stock plants, but root tissues and youngest leaves generally tested positive for the viroid. In addition, the pathogen was detected in the inflorescence tissues of symptomatic flowering plants of several cannabis genotypes by RT-PCR. A comparison of 11 cannabis genotypes, utilizing plants ranging in age from 6 weeks to 10 weeks following the initiation of rooting from cuttings, and assessing presence/absence of HLVd using a LAMP assay, demonstrated that root tissues consistently showed the highest frequency of detection in these plants compared to the leaves and petioles. These results were confirmed by RT-PCR, and also by RT-qPCR and ddPCR, all of which confirmed the presence of HLVd in different tissues of infected cannabis plants, including inflorescences, and demonstrated consistent viroid presence in root tissues. The LAMP (loop-mediated isothermal amplification) technique was developed for the detection of plant pathogens due to its speed, high specificity, sensitivity, efficiency, and isothermal conditions suitable for field conditions [47,48]. LAMP is a one-step amplification assay that amplifies the target DNA or RNA sequence and requires two or three pairs of primers to detect six distinct regions in the target sequence [48]. Additionally, the target gene fragment is usually short, producing a series of DNA fragments that are of different sizes [48,49]. In this study, LAMP was used to demonstrate the presence of HLVd in various tissues of cannabis plants (leaves, petioles, and roots). Some limitations of the LAMP technique include the high risk of cross-contamination, as well as carry-over contamination and off-target amplification, which subsequently can result in false-positives due to the high efficiency of DNA amplification in this method [50,51,52,53].

In a recent study, a combined method utilizing RT-LAMP with RT-qPCR was described for HLVd detection in cannabis plants (LAMP/qPCR) [54]. The results provided comparable results to standard RT-qPCR methods and confirmed HLVd was present in various tissues (roots, petioles, leaves) of plants varying in age from 5 weeks to 10 weeks at high levels. A sampling strategy based on leaf tissues taken from 5–10-week-old plants was recommended to be the most suitable approach for the early detection of HLVd. Our findings using RT-PCR, RT-qPCR, and ddPCR demonstrated that consistent and high levels of HLVd were present in root tissues of 6–10-week-old cannabis plants, as well as in 3–4-month-old stock plants. Leaf tissues sampled from these plants sometimes resulted in negative or low titers of HLVd, depending on the location of the samples. The results obtained from leaf samples also varied according to the genotype of the stock plant tested. The results from ddPCR showed that high levels of viroid genomes (>10,000 copies per reaction) were present in the roots of one genotype; in a second genotype, there were 10,000–100,000 copies in the roots and youngest leaves, indicating it was highly susceptible to infection. There were also significant differences in the levels to which the viroid accumulated in the leaves of flowering plants among eight cannabis genotypes grown adjacent to one another in the same environment, ranging from undetectable to >10,000 genomes, reflecting differences in susceptibility to infection or spread. The earliest detection of HLVd by RT-qPCR was observed in samples of root tissues from 2-week-old rooted cuttings in this study. Selection of the appropriate tissues, time of sampling, and cannabis genotype can influence the accuracy of the detection of HLVd. Currently, several commercial laboratories offering diagnostic services for HLVd testing have identified root tissues as the preferred sampling material based on the earlier, higher, and more consistent accumulation of viroid levels in plants ranging from 2 weeks to 3 months of age (https://tumigenomics.com/hop-latent-viroid-information, accessed on 10 October 2023, https://medicinalgenomics.com/hop-latent-viroid-in-cannabis/, accessed on 10 October 2023, https://3riversbiotech.com/3-rivers-biotech-identifies-root-tissue-from-mature-plants-as-the-most-reliable-to-detect-hop-latent-viroid-hlvd/, accessed on 10 October 2023).

In inflorescence tissues of cannabis genotype ‘Mac-1’, a highly susceptible genotype, HLVd was detected at the highest titer (>150,000 copies) within the central inflorescence tissues that had been stripped of the surrounding inflorescence leaves and the fan leaves. The viroid was also present at high levels (10,000 copies) in the inflorescence leaves, but not in the fan leaves. In adjacent asymptomatic plants, the viroid was absent in inflorescence tissues, but was present at low levels (five copies) in vegetative leaf tissue. When HLVd-infected ‘Mac1′ was used as a female parent and fertilized with pollen from another cannabis genotype, the resulting seeds had a high incidence of HLVd infection (70–87.5%) as determined by RT-PCR and RT-qPCR. The ddPCR method has been previously used for the quantitative assessment of a range of different plant pathogens [55,56,57,58,59,60]. The main principle of ddPCR, as in other PCR-based methods, including quantitative PCR (qPCR), is the specific amplification of a nucleic acid target. The distinctive feature of dPCR is the separation of the reaction mixture into thousands to millions of partitions, which is followed by a real-time or end-point detection in each partitioned reaction. The distribution of target sequences into partitions (droplets) is described by the Poisson distribution, thus allowing the accurate and absolute quantification of the target from the ratio of positive against all partitions at the end of the reaction. This omits the need to use reference materials with known target concentrations and increases the accuracy of quantification at low target concentrations compared to qPCR. The ddPCR has also shown higher resilience to inhibitors in a number of different types of samples. This is the first application of ddPCR to detect a pathogen in cannabis plants and the first demonstration of its use for the quantification of HLVd.

A comparison of the whole-genome sequences of HLVd from hop, hemp, and cannabis plants worldwide using phylogenetic analysis revealed a lack of diversity among the sequences included. Two single-nucleotide polymorphisms (SNPs) detected did not influence the overall alignment of the sequences, and all isolates were placed into one large group. In contrast to HLVd, a related viroid—potato spindle tuber viroid—shows considerably more sequence heterogeneity among isolates from different hosts and regions [61,62]. Continuous monitoring for any potential changes in the genome of HLVd will be needed to detect possible variants that may arise in the near future as the pathogen continues to spread and evolve. In BCTV, the evolution of new strains (biotypes) has been shown to occur in different regions where hemp and other crops are cultivated, and currently up to 11 strains have been identified [41]. In TSV, a variant found to be present in hemp plants had only 80–83% sequence similarity to previous strains infecting other crops [3]. Variation in the CasaMV1 sequences from hemp plants was also observed, with 88–99% nt identity to sequences from *C. sativa* [3]. The CYVaV sequences had 90% nt identity with CYVaV identified from citrus [3]. Therefore, there is some evidence for the potential evolution of new and genetically diverse virus/viroid strains that can infect and become established in hemp and cannabis crops. Whether the virulence patterns have been altered in these strains has not been established. Similarly, whether commonly encountered and widespread viruses on other crops, such as AMV, CMV, and TMV, will become problematic on cannabis and hemp crops remains to be seen. Continuing bioinformatics studies on populations of viruses and viroids that may be present in cannabis and hemp plants are needed. As well, the impact on host plant growth and development following pathogen infection and reproduction need to be established, preferably in studies involving artificial inoculations. Current reports of virus/viroid presence need to include further investigations into their potential role in causing symptoms.

An extensive review of the potential impact of viruses/viroid on cannabis and hemp (historical and current) has been recently published by Miotti et al. [28]. Some potential aphid-transmissible viruses that can infect cannabis and hemp plants under artificial inoculation conditions, but have not yet become widespread under commercial conditions, include AMV, CMV, and PVY [28,63]. Viruses vectored by thrips which can also pose a threat include Tobacco streak virus and Tomato spotted wilt virus [28]. Many of these emerging viruses are potentially also seed-borne [28]. These observations suggest that cannabis and hemp crops are susceptible to a range of viruses, and that the insect-vectored pathogens appear to have the greatest potential to cause damage under widespread cultivation conditions, with mechanically transmitted pathogens less so. The exception to this is HLVd, which is mechanically transmitted and is presently widespread on cannabis [21]. Therefore, the development of diagnostic assays that can be applied in seed-testing programs will be important for the cannabis and hemp industries moving forward. Such testing programs are currently unavailable in many production areas. In the present study, HLVd presence on cannabis and hemp seeds was confirmed by RT-PCR. Given the high titer of viroid present in cannabis inflorescence tissues (which consist of clusters of pistils) of a susceptible genotype, fertilization of the ovules contained in these infected tissues by pollen originating from a male plant would likely yield a high frequency of seed-borne transmission, which was demonstrated in this study for HLVd. Infection during seed development may cause seed abortion and a reduction in the seed weight of surviving seeds. While it was not determined whether HLVd was present on the seed coat or internally within the seed, it is likely to be both. The viroid was also detected on cannabis seeds that had been stored for more than 2 years (dating back to 2021), suggesting that seed stocks of cannabis should be tested prior to widespread distribution. The viroid levels differed significantly from one seed to the next, even within the same batch of seeds. This variability should be considered in seed-testing protocols or when establishing levels of transmission of the viroid to seedlings, since the resulting levels of the viroid may be variable. Seeds of hemp infected with HLVd gave rise to infected seedlings at a high frequency in the present study, but the symptomology on these plants has not been established. The diagnostic approaches described in this study should aid in the routine screening of plants and seeds for the range of pathogens currently reported to affect cannabis and hemp crops.

A comparison of the prevalence of fungal and oomycete pathogens affecting cannabis and hemp crops to the viruses/viroid disease complex indicates that, while there are new reports of the occurrence of the former group of pathogens on these hosts in different geographic regions, there is no evidence for selection of new pathogen strains that are adapted to cannabis and hemp, i.e., the pathogens are shown to have originated and spread from previous crops or adjacent crops [1]. A similar situation may be taking place with the virus/viroid pathogens, but preliminary evidence may suggest an evolving suite of these pathogens may be found infecting these crops in the future. The applications of the molecular diagnostic and bioinformatics methods described in this study should provide useful information to address the evolving challenges facing cannabis and hemp crops as a result of these evolving multiple and assorted pathogens, which could become a limiting factor to production in certain regions [64].

## 4. Materials and Methods

### 4.1. Detection of Fungal and Oomycete Pathogens on Cannabis

#### 4.1.1. Symptoms and Pathogen Isolation

Cannabis plants displaying symptoms that included stunting and yellowing of vegetative and flowering plants and showed internal stem discoloration (Figure 1a–c), as well as visible evidence of fungal growth on the inflorescences (Figure 1d–f), were obtained from indoor production sites and greenhouses in which the cultivation of a range of different genotypes (strains) occurred. To recover fungi and oomycetes from the roots, stems, and inflorescences, tissue pieces measuring 0.5 mm^2^ were taken from symptomatic plants and surface-sterilized by immersing them in 10% bleach (0.525% NaOCl) for 30 s, followed by 70% EtOH for 30 s, and three rinses in sterile distilled water. The pieces were transferred to potato dextrose agar containing 140 mg/L streptomycin sulfate and incubated at room temperature (22–25 °C) for 5–7 days. Emerging colonies were subcultured onto fresh agar medium. Morphological criteria that included colony features and spore characteristics were used to tentatively assign a genus and species identification to the various colonies that were recovered. These cultures were then used to conduct the molecular analysis, as described below. Symptomatic plant tissues were also used directly for DNA extraction, as described below, and used for PCR. DNA from noninfected cannabis tissues was included as a control.

#### 4.1.2. PCR Analysis

DNA was extracted from 50–100 mg of active growing mycelium or from 50 mg of plant tissues using the QIAGEN DNeasy Plant Mini Kit (Hilden, Germany) (cat. no. 69104). A pair of universal eukaryotic primers that amplified the ribosomal DNA in the internal transcribed region (ITS1-5.8S-ITS2) was used to amplify each sample. The primers (UN-UP18S42 (5′-CGTAACAAGGTTTCCGTAGGTGAAC-3′) and UN-LO28S576B (5′-GTTTCTTTTCCTCCGCTTATTAATATG-3′) [4] were used with the following PCR conditions: initial denaturation at 94 °C for 3 min, 40 cycles of denaturation at 94 °C for 30 s, annealing at 55 °C for 45 s, extension at 72 °C for 2 min, and a final extension at 72 °C for 7 min, followed by a 4 °C hold. PCR bands were cut from the gel, collected using the MinElute Gel Extraction Kit (Valencia, CA, USA) (cat. no. 28604), and 8 uL was sent to Eurofins Genomics (https://www.eurofinsgenomics.com/en/home/, accessed on 1 October 2023) for sequencing. The resulting sequences were compared to ITS1-5.8S-ITS2 sequences from the National Center for Biotechnology Information (NCBI) GenBank database using BLAST [65] to confirm species identity, using cut-off values >99%. Representative sequences were deposited in GenBank.

### 4.2. Detection of Viral Pathogens on Cannabis

#### 4.2.1. Symptoms

Leaves of cannabis plants displaying foliar symptoms of mosaic, mottling, chlorosis, and line patterns that resembled those caused by viral infection (Figure 4) were collected from several genotypes. These symptoms were observed on vegetative and flowering plants of genotypes ‘OG Kush’ (OG), ‘Headband’ (HB), ‘Motor Breath’ (MB), and ‘Golden Papaya’ (GP). Leaf tissues from these symptomatic plants were subjected to several diagnostic approaches, as described below, to determine if the observed symptoms observed were caused by viruses. The samples were collected on various times during 2021–2022.

#### 4.2.2. PCR with Broad-Spectrum and Specific Primer Sets

Leaf tissues from the plants of genotypes OG and HB were used for the total RNA extraction using an RNeasy Plant Mini Kit (Qiagen Sciences, Germantown, MD, USA). A small portion of the extracted RNA was converted to cDNA using the two-step RT-PCR system, iScriptTM Select cDNA Synthesis Kit (Bio-Rad Laboratories Ltd., Mississauga, ON, Canada). PCR was conducted on the cDNA samples generated using individual degenerate broad-spectrum primer sets under the conditions described in Appendix A. These primer sets were designed to detect viruses belonging to one of the following genera: Tobamovirus, Nepovirus, Potyvirus, and Ilarvirus, or to the species *Turnip ringspot virus* (Comovirus), *Alfalfa mosaic virus* (Alfamovirus), *Tobacco mosaic virus* (Tobamovirus), and *Cucumber mosaic virus* (Cucumovirus). Control tobacco tissues (*Nicotiana tabacum* cv. ‘Samsun’) infected with the tobacco mosaic virus U1 strain and alfalfa mosaic virus were also included (provided by the Canadian Plant Virus Collection at Agriculture and Agri-Food Canada). Any amplified PCR products of the expected size for the specific primer set used were then purified with the MinElute PCR Purification Kit (Qiagen Sciences, Germantown, MD, USA) and sent to Eurofins Genomics (https://www.eurofinsgenomics.com/en/home/, accessed on 1 October 2023) for sequencing.

#### 4.2.3. Transmission Electron Microscopy

Sap samples from symptomatic leaf tissues of the cannabis genotype MB, as well as from *N. tabacum* with symptoms of Tobacco mosaic virus previously inoculated with a confirmed TMV strain (U1 strain) (provided by the Canadian Plant Virus Collection at Agriculture and Agri-Food Canada), were obtained by grinding leaves with a mortar and pestle. The preparation method for the crude leaf extracts was performed as per Hitchborn and Hills [66]. The sample was observed with a Hitachi H-7100 Transmission Electron Microscope (TEM) (100 kv) and imaged in Gatan Digital Micrograph software (v. 2.31.734.0; Gatan Inc., Pleasanton, CA, USA). Virions were searched for systematically from top to bottom, from left to right. Preliminary searches were done at 5000–6000× magnification, and increased to 30,000–40,000× magnification for the imaging and measurement of individual virions. Fifty virions (if present) were imaged from each host sample, and the length and diameter of each was measured in the Gatan Digital Micrograph software.

#### 4.2.4. Host Range Studies

A mechanical transmission study utilizing select plant species was conducted by sap inoculations using symptomatic cannabis leaf tissues from genotypes OG and HB (Figure 4) as source inoculum. The following plant species were included: *Nicotiana clevelandii* (Cleveland’s tobacco), *N. glutinosa* (Peruvian tobacco), *N. tabacum* ‘Samsun’ (cultivated tobacco), *Chenopodium quinoa* (quinoa), *C. amaranticolor* (goosefoot), *Gomphrena globosa* (globe amaranth) (all seeds were provided by Agriculture and Agri-Food Canada), *Solanum lycopersicoides* (tomato), *Cucumis sativus* (cucumber) (seeds were purchased from West Coast Seeds), and *Urtica diocea* (stinging nettle plants originating from rhizomes collected from an outdoor forested location). Seeds of all plants (except *Urtica diocea*) were planted in a cocofibre:perlite potting medium (3:1) and placed under a 24 h photoperiod for 3 weeks, after which they were transplanted into individual pots and fertilized with a 20:20:20 (N:P:K) fertilizer. One week later, the plants were transferred to complete darkness for 24 hr prior to inoculation. Leaves of all plants were dusted with 320 grit carborundum (Thermo Fisher Scientific, Waltham, MA, USA) and gently rubbed with a leaf slurry prepared by grinding 0.5–1 g of tissue from each genotype in 1–2 mL 50 mM sodium phosphate buffer (PO_4_) at a pH of 7.5. Inoculated and control plants (*n* = 3–5 for each set of inoculated or control plants per species) were maintained under a 12:12 hr photoperiod and observed daily for symptoms for up to 3 weeks (Figure 6).

#### 4.2.5. High-throughput Sequencing (HTS)

Leaves from cannabis genotypes OG and HB were taken from plants displaying symptoms of mottling and mosaic (Figure 4), as well from plants of 6 additional genotypes, many of which exhibited stunting and reduced inflorescence growth (Figure 7), for HTS. These genotypes were ‘Powdered Donuts’ (PD), ‘Mac-1’ (Mac), ‘Pink Kush’ (PK), ‘Black Cherry’ (BC), CBD, and G54-2. Approximately 1 g of leaf tissue was used for virus and viroid double-stranded RNA (dsRNA) extraction. Extraction of dsRNA, removal of genomic DNA and single-stranded RNA (ssRNA), and construction of cDNA libraries were performed as described by Su et al. [67]. The cDNA libraries were paired-end sequenced on a HiSeq 4000 platform (Illumina) by Applied Biological Materials Inc. (Richmond, BC, Canada). Sequence reads were trimmed to remove low-quality reads and the adaptor sequences, and assembled using the de novo assembly algorithm of the CLC Genomics Workbench v20 (Qiagen Sciences, Germantown, MD, USA). Assembled sequences were compared to a database of known viruses derived from the National Center for Biotechnology Information (NCBI) GenBank database.

### 4.3. Molecular Assays for Viroid and Other Viral Pathogens

#### 4.3.1. RNA Extractions and cDNA Preparation

In addition to the virus-like symptoms shown in Figure 4, cannabis plants representing eight genotypes with symptoms of stunting, reduced stem growth, and poor development of the inflorescences (Figure 7) were subjected to molecular analysis. These genotypes were PD, Mac, PK, BC, CBD, G54-2, HB, and OG. Leaf and flower tissues were subjected to RT-PCR using primers that were developed as described below (see Table 6). For total RNA extractions, fresh leaf samples were placed in paper bags and freeze-dried using a Genesis 25 Freeze Drier (SP Industries Inc., Gardiner, NY, USA), after which the leaves were gently pulverized to create a semihomogenous crumble mix inside the bags. Approximately 500 mg of tissue was transferred to universal extraction bags (Bioreba AG, Reinach, Switzerland) and suspended in 5 mL of UltraPure water (Thermo Fisher Scientific, Waltham, MA, USA). The samples were ground using a HOMEX 6 homogenizer (Bioreba AG) and homogenate saps were transferred to 1.5 mL tubes using disposable transfer pipettes. From these tubes, 100 µL of homogenate sap was transferred to 2.0 mL tubes containing 300 µL of RB buffer (Omega Bio-tek, Norcross, GA, USA) with fresh 0.21% (*v*/*v*) β-mercaptoethanol (Thermo Fisher Scientific). Samples were loaded onto a QIAcube Connect (Qiagen, Germantown, MD, USA) for RNA extraction using an E.Z.N.A.^®^ Plant RNA Kit (Omega Bio-tek) and the QIAcube program for the RNeasy Plant Mini Kit (Qiagen, Germantown, MD, USA). The elution volume was set to 50 µL and the eluted RNA was stored at −20 °C. RNA concentrations and purity were determined using a NanoDrop 1000 spectrophotometer (Thermo Fisher Scientific) (Appendix A). The cDNA was generated using a SuperScript™ VILO™ Master Mix kit (Thermo Fisher Scientific) and 500 ng of template RNA per reaction. The following thermal cycling protocol was used on a T100™ Thermal Cycler (Bio-Rad, Hecules, CA, USA): lid temperature set at 105 °C; primer binding at 25 °C for 10 min; reverse transcription at 50 °C for 10 min; reaction termination at 85 °C for 5 min, then held at 10 °C until storage at −20 °C.

#### 4.3.2. Primer Design for RT-PCR

Using the HTS data from cannabis leaf samples of genotypes HB and OG, primers were either designed from this study (specifically for hop latent viroid (HLVd) and CasaMV1 mitovirus) or from previously published reports (Table 6). Primers were designed using Primer-BLAST [68] and specificity was tested against all plant, virus, bacteria, and fungal sequences in the nonredundant nucleotide collection (nr) from the National Center for Biotechnology Information (NCBI) GenBank database. Sequencing primer specificity was checked to ensure nonspecifics were not generated. To achieve coverage of the primed regions, two sequencing primer sets were evaluated: HLVd seq F1/HLVd seq R1 and HLVd quant F1/HLVd seq R2. Specificity was checked by PCR of 1 µL of generated cDNA in 20 µL reactions of the Q5^®^ High-Fidelity 2X Master Mix (New England Biobabs, Ipswich, MA, USA) with sequencing primers at final concentrations of 500 nM. The following thermal cycling protocol was used: lid temp set at 105 °C, initial denaturation at 98 °C for 30 s; 30 cycles of 98 °C; denaturation for 10 s; 58 °C annealing for 30 s; 72 °C elongation for 80 s; a final elongation at 72 °C for 7 min, and held at 10 °C until stored at −20 °C. Following PCR, these were electrophoresed in 2% agarose (Thermo Fisher Scientific) at 100 V for 60 min in 0.5x TBE buffer (Thermo Fisher Scientific). The agarose gel was poststained in 1x SYBR™ Safe DNA Gel Stain (Thermo Fisher Scientific) in 0.5x TBE (Thermo Fisher Scientific) for 30 min. The stained gel was imaged using a ChemiDoc MP imaging system (Bio-Rad, Hercules, CA, USA). In the case of both primer sets, nonspecific bands were not observed, although multiple bands (multimers) were seen due to the circular nature and small size of the HLVd genome.

**Table 6 ijms-25-00014-t006:** Selected HLVd primers used for this study.

Target	Primer Name	Sequence (5′–3′)	Source
HLVd	HLVd seq Forward1	ATACAACTCTTGAGCGCCGA	Eastwell and Nelson [69]
	HLVd seq Reverse1	CCACCGGGTAGTTCCCAACT	Eastwell and Nelson [69]
	HLVd seq Reverse2	AGGACGCGAACAAGAAGAAG	This work
	HLVd quant Forward1	GTTGCTTCGGCTTCTTCTTG	This work
	HLVd quant Reverse1	AGTTGTATCCACCGGGTAGT	This work
Cannabis EF1*α*	Cannabis *EF1α* Forward	TGTTTTGCACGGATCAGTTTG	Guo [70]
	Cannabis *EF1α* Reverse	AATGCCGACCGCTACAGTTC	Guo [70]

### 4.4. Sequence Diversity and Molecular Phylogeny of Hop Latent Viroid

The evolutionary history was inferred by using the maximum likelihood method and Kimura 2-parameter model [71]. The bootstrap consensus tree inferred from 1000 replicates [72] was taken to represent the evolutionary history of the taxa analyzed. Branches corresponding to partitions reproduced in less than 70% bootstrap replicates were collapsed. The percentage of replicate trees in which the associated taxa clustered together in the bootstrap test (1000 replicates) are shown next to the branches [72]. Initial tree(s) for the heuristic search were obtained automatically by applying Neighbor-Join and BioNJ algorithms to a matrix of pairwise distances estimated using the maximum composite likelihood (MCL) approach, and then selecting the topology with superior log likelihood value. This analysis involved 21 nucleotide sequences from GenBank. Only full-length sequences were used, and selections were chosen to represent wide geographic regions. All positions with less than 95% site coverage were eliminated, i.e., fewer than 5% alignment gaps, missing data, and ambiguous bases were allowed at any position (partial deletion option). There was a total of 185 positions in the final dataset. Evolutionary analyses were conducted in MEGA11 [73].

### 4.5. Droplet Digital PCR

Quantitative primer specificity was also tested using the primer sets HLVd quant F1/HLVd quant F2 and Cannabis *EF1α* F/Cannabis *EF1α* R (Table 6). Specificity was evaluated using only the cDNA generated (see above) from the RNA of one cannabis genotype (PD). PCR was conducted using 20 µL reactions of QX200™ ddPCR™ EvaGreen Supermix (Bio-rad) with quantitative primers at final concentrations of 150 nM, 3 µL of 10^−2^ diluted PD cDNA template per reaction, and without generating droplets. The PCRs were conducted across a 5-point thermal gradient ranging from 55.8 °C to 62 °C to determine the ideal annealing temperature. Following PCR, these were electrophoresed, stained, and imaged, as described above. Neither primer set produced nonspecific bands, and both sets performed equally well across the thermal gradient. A temperature of 58 °C was selected for the subsequent ddPCR, as higher temperatures can produce inadequate separation between positive and negative droplets, termed ‘rain’. A total of five cannabis genotypes were tested by ddPCR, and the levels of HLVd present were quantified.

### 4.6. Multiplex Taqman RT-qPCR

Tissues were obtained from the flowering plants of genotypes PD and Mac showing symptoms, as seen in Figure 7. Approximately 100 mg of fresh inflorescence tissues were placed directly into 750 uL of RNAlater^®^ solution (Thermo Fisher Scientific) in a 2 mL screw-cap tube and placed on ice. The samples were either processed the same day or stored overnight at 4 °C and processed the following day. Total nucleic acids were extracted as described by Mark et al. [74], with slight modifications. The RNAlater^®^ was removed from the 2 mL tube followed by adding 2.3 mm diameter zirconia–silica beads and 1.0 mm diameter glass beads (BioSpec, Bartlesville, OK, USA). The tissue was then macerated using a TissueLyser (Qiagen) at 25 Hz for 1 min. Once homogenized, 1 mL of extraction buffer (2% CTAB, 2% PVP40,000, 25 mM EDTA at a pH of 8, 100 mM Tris-HCl at a pH of 8, and 2.5 mM NaCl) prewarmed at 65 °C was added to each sample and incubated at 65 °C for 10 min. The homogenate was centrifuged at max speed (>16,000× *g*) for 5 min at 4 °C and the supernatant transferred to a new sterile 1.5 mL Eppendorf tube. An equal volume (750 µL) of chloroform isoamyl alcohol (24:1) was added and centrifuged at max speed (>16,000× *g*) for 5 min. The supernatant (550 uL) was transferred to a new 1.5 mL Eppendorf tube, and 0.6 volumes of cold isopropanol were added and centrifuged again at max speed (>16,000× *g*) for 5 min. Isopropanol was decanted and about 650 µL of 70% ethanol was added onto the pellets, centrifuged at max speed (>16,000× *g*) for 3 min, and air-dried before resuspending into 40 µL of nuclease-free water. RT-qPCR was performed on a Bio-Rad CFX96 instrument using TaqPath™ 1-Step Multiplex Master Mix (No ROX) (Thermo Fisher Scientific). The reagents were brought to room temperature prior to prevent infrequent nonspecific signal increases found when master mixes are prepared on ice. Each 20 μL reaction mixture contained 5 μL of 4 × TaqPath 1-Step Multiplex Master Mix (No ROX; Thermo Fisher Scientific), 1 μL of primer/probes mix, and 4 μL of extracted total nucleic acid. The primers used are shown in Table 7. The final concentrations of primers and probes were 0.3 μM (target primers), 0.05 μM (target probes), 0.15 μM (internal control primer), and 0.05 μM (internal control probe). The cycling program on a Bio-Rad CFX96 instrument was as follows: uracil–DNA glycosylase incubation at 25 °C for 2 min; reverse transcription at 53 °C for 10 min; polymerase activation at 95 °C for 2 min; 40 cycles of PCR at 95 °C for 3 s and 60 °C for 30 s (signal acquisition). The filter combinations were 465–510 (FAM; *UBQ* P), 540–580 (HEX; HLVd P1), and 610–670 (Cy5; HLVd P2).

### 4.7. LAMP Assays

Tissues were sampled from plants showing symptoms, as illustrated in Figure 4, as well as from asymptomatic plants (showing no visible alteration of growth). Leaves, petioles, and roots were collected from 11 cannabis genotypes for comparison. Plants ranged in age from 6 to 10 weeks after the initiation of cuttings. Approximately 100 mg of fresh tissues were placed directly into 750 uL of RNAlater^®^ solution (Thermo Fisher Scientific) in a 2 mL screw-cap tube and placed on ice. The samples were either processed the same day or stored overnight at 4 °C and processed the following day. Total nucleic acids were extracted as described by Mark et al. [74], with slight modifications. The RNAlater^®^ was removed from the 2 mL tube followed by adding 2.3 mm diameter zirconia–silica beads and 1.0 mm diameter glass beads (BioSpec, Bartlesville, OK, USA). The tissue was then macerated using a TissueLyser (Qiagen, Germantown, MD, USA) at 25 Hz for 1 min. Once homogenized, 1 mL of extraction buffer (2% CTAB, 2% PVP40,000, 25 mM EDTA at a pH of 8, 100 mM Tris-HCl at a pH of 8, and 2.5 mM NaCl) prewarmed at 65 °C was added to each sample and incubated at 65 °C for 10 min. The homogenate was centrifuged at max speed (>16,000× *g*) for 5 min at 4 °C and the supernatant transferred to a new sterile 1.5 mL Eppendorf tube. An equal volume (750 µL) of chloroform isoamyl alcohol (24:1) was added and centrifuged at max speed (>16,000× *g*) for 5 min. The supernatant (550 uL) was transferred to a new 1.5 mL Eppendorf tube, and 0.6 volumes of cold isopropanol were added and centrifuged again at max speed (>16,000× *g*) for 5 min. Isopropanol was decanted, and about 650 µL of 70% ethanol was added onto the pellets, centrifuged at max speed (>16,000× *g*) for 3 min, and air-dried before resuspending into 40 µL of nuclease-free water. RT-LAMP primer sequences were designed using New England Biolabs^®^ (NEB) LAMP primer design tool and synthesized by Integrated DNA Technologies (Coraville, IA, USA) (Table 8). RT-LAMP reactions were performed using WarmStart^®^ Fluorescent LAMP/RT-LAMP Kit (with UDG) (New England Biolabs, Whitby, ON, Canada; E1708) with standard primer concentrations (0.2 μM F3, 0.2 μM B3, 1.6 μM FIP, 1.6 μM BIP, 0.4 μM Loop F, 0.4 μM Loop B) in 25 μL on 96-well plates at 65 °C for 30 min in a Bio-Rad CFX96 instrument.

### 4.8. Monitoring Distribution of HLVd in Various Plant Tissues

#### 4.8.1. Distribution within Stock Plants

To assess how specific diagnostic assays can be used to monitor the presence of HLVd in different tissues of cannabis plants, RT-PCR with primers HLVd seq F1/HLVd seq R1 were used according to the conditions specified above (Section 4.3.2). Stock (mother) plants of several genotypes that were 3–4 months old were sampled at different positions on the plant—leaves were collected from the top (youngest growth), middle, and bottom (oldest growth), as well as from roots. The plants did not display any obvious symptoms of disease, such as stunting or leaf curl (Figure 14). Each leaf sample (approximately 5 gm fresh weight) was frozen at −80 C until used. The extractions were conducted using the Qiagen RNeasy Plant Mini Kit (cat. #74904) according to the manufacturer’s instructions. The final RNA product was eluted with 52 µL nuclease-free H_2_O. The QIAGEN OneStep RT-PCR Kit (cat. #210212) was used for reverse transcription and PCR amplification. The reaction mixture contained 14 µL of water, 5 µL of 5x reaction buffer, 1 µL of dNTPs (10 mM), 1.5 µL each of primers HLVd seq F1/HLVd seq R1, 1 µL of RNA template, and 1 µL of enzyme mix, resulting in a total volume of 25 µL. All PCR amplifications were performed in a MyCycler thermocycler (BIORAD) with the following program: 30 min at 50 °C, 15 min at 95 °C, followed by 35 cycles of 30 s at 94 °C, 30 s at 58 °C, 60 s at 72 °C, and final extension at 72 °C for 10 min. The resulting PCR products were run on a 1% TAE agarose gel, and images were captured with E-gel imager (Life Technologies, Carlsbad, CA, USA). Bands of the expected size (ca. 256 and 512 bp) were purified with QIAquick Gel Extraction Kit and sent to Eurofins Genomics (Eurofins MWG Operon LLC 2016, Louisville, KY, USA) for sequencing. The resulting sequences were compared to the corresponding HpLVd sequences from the National Centre for Biotechnology Information (NCBI) GenBank database to confirm identity. In addition, primers HLVd quant F1/HLVd quant F2 were also used to quantify viroid levels in the same tissues, as described in Section 4.5 above.

#### 4.8.2. Distribution within Inflorescence Tissues

Mature inflorescences were obtained at harvest from a plant (approximately 12 weeks of age that had been grown under a photoperiod of 12:12 hr to induce flowering for 8 weeks) of genotype ‘Mac-1’ which was previously confirmed to be infected by HLVd. The leaves surrounding the inflorescence that included large fan leaves and smaller inflorescence leaves (Figure 15) were manually dissected from the inflorescence using a scalpel and used for RNA extraction, followed by RT-PCR and ddPCR to compare the relative viroid concentrations in these tissues relative to that present in the whole inflorescence (Figure 15).

#### 4.8.3. Distribution of HLVd within Cuttings from Infected Stock Plants

Vegetative cuttings were taken directly from a stock plant of genotype ‘Blue Dream’ confirmed to be infected with HLVd and were rooted and tested for HLVd presence using RT-PCR and RT-qPCR after 14 days. The conditions for rooting are described elsewhere [4]. A total of 7 cuttings were subjected to RT-PCR, and 4 of the cuttings were also analyzed by RT-qPCR, and the results were compared.

#### 4.8.4. Presence of Pathogens in Fresh and Dried Cannabis Inflorescences

Following harvest of inflorescences from a number of genotypes of cannabis plants, both fresh and dried flowers were tested for the presence of specific pathogens by PCR with the universal primers for fungal/oomycete pathogens, and by RT-PCR for HLVd. The fresh flowers were tested immediately after they were obtained by removing inflorescence leaves and subjecting them to PCR, as described previously (Section 4.1.2). For dried samples, flowers were subjected to commercial drying conditions (5 days at 20–22 °C and 50–55% relative humidity), after which the tissue samples were obtained and subjected to PCR with HLVd primers, as described in Section 4.3.2. Up to 20 fresh and dried samples were included in the analysis. Resulting bands observed in these analyses following PCR were collected and sent for sequencing to confirm which pathogens were present.

#### 4.8.5. Detection of HLVd and Mitovirus in Cannabis Seed

A cross was made between an infected ‘Mac1′ female plant and pollen from a healthy male plant of genotype ‘GPie’ by collecting pollen and manually transferring to the stigmatic surfaces. The plants were grown in isolation after crossing for 10 weeks, after which seeds that had developed were collected and frozen at −80 C. Individual seeds were ground and the RNA was extracted, as previously described, and subjected to RT-PCR using HLVd primers and CasaMV1 primers.

### 4.9. Detection of Beet Curly Top Virus

Cannabis plants with symptoms of stunting, extensive curling of young leaves, and twisted and deformed stem growth (Figure 19) were used. DNA from approximately 100 mg of stem tissues from these plants was extracted with a Qiagen DNeasy extraction kit and eluted into 100 uL of elution buffer. The extracted DNA was then diluted 1:10 in TE buffer before PCR. The PCR was carried out in 20 uL reactions using Thermo Fisher DreamTaq according to the manufacturer’s instructions. Primer sequences used are shown in Table 9. The PCR conditions were 94 °C for 5 min followed by 40 cycles of 94 °C for 30 s, 58 °C for 60 s, 72 °C for 90 s, and a final extension of 72 °C for 10 min. The PCR products were run on a 2% TAE agarose gel at 100 V for 30 min before imaging.

### 4.10. Diagnostic Approaches to Detect Viral and Viroid Pathogens Affecting Low THC-Containing Cannabis sativa L. (hemp)

#### 4.10.1. Next-Generation Sequencing (NGS)

Leaf samples were obtained from several outdoor hemp fields in Colorado during 2019, 2021, and 2022, and subjected to shotgun metagenomic analysis, as described by Chiginsky et al. [3]. In most cases, samples were taken from symptomatic plants, as shown in Figure 21. In each year, total RNA was extracted from a composite of 3–5 leaves from outdoor hemp-production fields in Colorado. Approximately 100 mg of leaf tissue samples was placed in a 2 mL microcentrifuge tube and stored at −20 °C until nucleic acid extraction was performed. Briefly, total RNA was extracted using the Qiagen Plant RNeasy kit and checked for the concentration using a Nanodrop One spectrophotometer (Thermo Fisher Scientific), and for quality using a Qubit 3.0 fluorometer (Thermo Fisher Scientific). Approximately 2 μg of RNA was submitted to the Colorado State University Next Generation Sequencing Facility, where library preparation, quality measurements, and sequencing were performed. Briefly, RNA quality was confirmed using an Agilent Tapestation instrument. Shotgun RNA libraries were constructed using the Kapa Biosystems RNA HyperPrep kit (Roche, IN, USA) according to the manufacturer’s instructions. Pooled libraries were sequenced on an Illumina NextSeq 500 instrument (Baltimore, MD, USA) to produce single-end 150 nucleotide (nt) reads. Bioinformatic analysis for the 2019 hemp samples are described in Chiginsky et al. [3]. Bioinformatic analysis for the 2021 and 2022 hemp samples was performed using CLC Genomics Work Bench (Qiagen). Reads were mapped to the hemp reference genome (assembly accession GCF_900626175.1). Remaining nonhost reads were assembled through the de novo assembly algorithm from the unmapped reads. Contigs and nonassembling reads were taxonomically categorized first by nucleotide-level alignment to the NCBI nucleotide (nt) database using Blastn, and then by protein-level alignment to the NCBI protein (nr) database. This produced a comprehensive classification of all nonhost reads. Candidate virus sequences were manually validated by aligning reads to draft genome sequences using bowtie2. Lastly, the raw sequence data were deposited in the NCBI Sequence Read Archive (SRA).

#### 4.10.2. RT-PCR with Specific Primer Sets

The universal BCTV primers, BCTV2- F, and BCTV2-R were used to amplify a 496 bp fragment of the coat protein (CP) gene, a region that is conserved among BCTV strains [41]. The amplification cycle consisted of a 94 °C initial denaturation for 5 min, 25 cycles of denaturation at 94 °C for 1 min, 58 °C annealing for 2 min, and 72 °C extension for 2 min, followed by a 10 min final extension. The GoTaq Flexi DNA polymerase (Promega, Madison, WI, USA) was used. All PCR products were visualized on 1% agarose gel. The PCR products were excised from the agarose gels and purified using the DNA Clean and Concentrator™-5 (Zymo Research, Irvine, CA, USA). One to two PCR products were randomly selected and submitted for Sanger sequencing at Genewiz Inc. to confirm the virus identity. The sequences for each BCTV strain were checked for identity against the nonredundant (nr) database using Blastn in the NCBI database. To confirm the presence of low-percentage nucleotide (nt)-identity viruses (<90%), 1 μg of total RNA was used to synthesize the cDNA using the Verso cDNA synthesis kit (Thermo Fisher Scientific) according to manufacturer’s instructions. For additional viruses, including Cannabis sativa mitovirus (CasaMV1), citrus yellow-vein-associated virus (CYVaV), and tobacco streak virus (TSV), the primers used are shown in Table 10. The RT-PCR was performed using GoTaq^R^ Flexi DNA polymerase (Promega, Madison, WI, USA). The amplification cycle consisted of 2 min at 95 °C, 40 cycles of 30 s at 95 °C, 30 s at 55 °C, and 35 s at 72 °C, followed by 5 min at 72 °C for all viruses except for citrus yellow-vein-associated virus (CYVaV), which had Tm of 51 °C.

#### 4.10.3. Sequence Diversity and Molecular Phylogeny of Beet Curly Top Virus

All currently available nucleotide sequences of BCTV capsid proteins (BCTV-CP) were retrieved from the GenBank database, after which their strains and sources of origins (hosts where viruses were identified) were determined according to the featured information in GenBank. Sequences representing each of the 11 BCTV strains were selected based on Strausbaugh et al. [41], and included 10 California/Logan, 8 Colorado, 2 Kimberly1, 10 Leafhopper71, 5 Mild, 10 Severe, 8 Worland, 6 Spanish curly top, 1 Pepper yellow dwarf, 2 Pepper curly top, and 1 Severe pepper. A total of 35 BCTV-CP sequences were found to be identified from *Cannabis sativa*, and all were included in the phylogenetic analysis. Multiple sequence alignments (total 98 sequences) were performed using MAFFT v7.505 [79], and the poorly aligned regions were trimmed using TrimAI v1.2.59 [80]. The phylogenetic tree was constructed using the maximum likelihood method implemented in the RAxML v8.2.12 with the GTR+G+I model for nucleotide substitution through the CIPRES Science Gateway Environment [81]. The robustness of each internal branch was estimated with 1000 bootstrap replications. The phylogenetic tree was visualized using FigTree v1.4.4 [82].

#### 4.10.4. Detection of HLVd on Hemp Seeds and on Thrips

A sample of hemp seeds from a commercial supplier was frozen at −80 °C, and then individual seeds were used to extract total RNA, as described in Section 4.8.1 above. In addition, a sample containing 8 adult thrips found feeding on the leaves of plants derived from these seeds at the true leaf stage were collected using a pair of forceps, and also frozen at −80 °C prior to analysis.

## Figures and Tables

**Figure 1 ijms-25-00014-f001:**
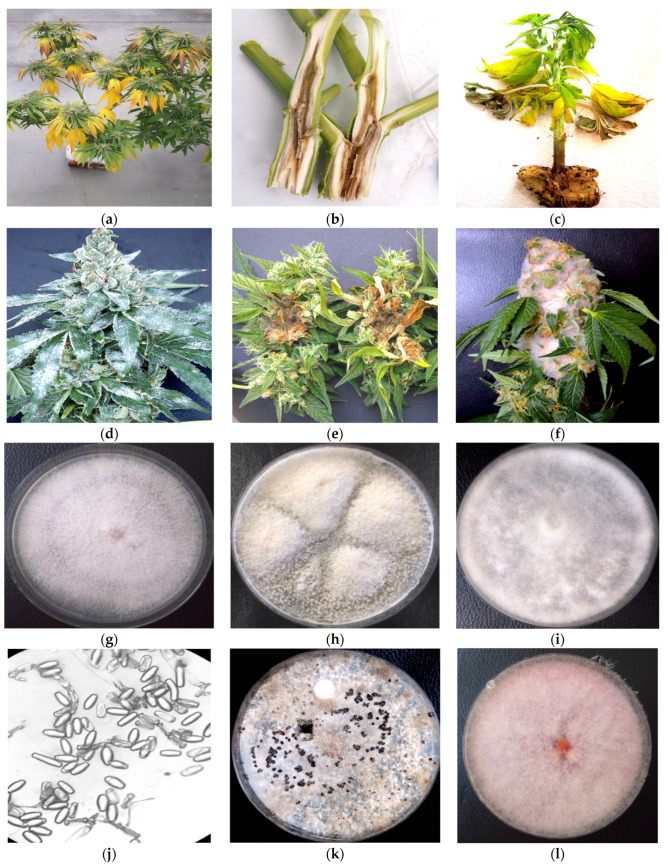
Symptomatic cannabis plants that were included in this study for molecular diagnostics. The plant tissues were used in PCR analysis, and isolations of potential pathogens were subsequently made on agar medium and then identified by PCR. The results indicated the presence of the following pathogens causing symptoms: (**a**) Yellowing of the plants caused by *Fusarium oxysporum*; (**b**) Stems with internal discoloration caused by *F. proliferatum*; (**c**) Plants with rotted roots and stem cankers due to *Pythium myriotylum*; (**d**) Plants with visible powdery mildew infections (*Golovinomyces ambrosiae*) on leaves and inflorescences; (**e**) Bud rot symptoms on inflorescences caused by *Botrytis cinerea*; (**f**) Pinkish-white mycelium growing over the inflorescence tissues caused by *Fusarium sporotrichiodes*. The corresponding pathogens that were isolated in culture and identified were: (**g**) *F. oxysporum*; (**h**) *F. proliferatum*; (**i**) *P. myriotylum*; (**j**) *G. ambrosiae*; (**k**) *B. cinerea*; (**l**) *F. sporotrichiodes*.

**Figure 2 ijms-25-00014-f002:**
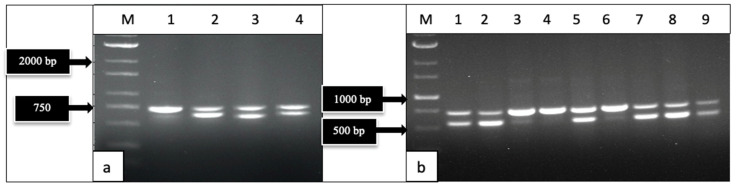
PCR with universal eukaryotic primers for the ITS1-5.8S-ITS2 region of ribosomal DNA from diseased and healthy cannabis tissues. (**a**) Tissues infected with powdery mildew (*G. ambrosiae*), as shown in Figure 1d. Lane 1 = noninfected tissues; lanes 2–4 = tissues with powdery mildew. A doublet banding pattern reflects pathogen-infected tissues. (**b**) Tissues infected with *Botrytis cinerea*, as shown in Figure 1e. Lanes 1, 2 = infected inflorescence leaves; lanes 3, 4, 6 = healthy leaves; 5 = infected ovary tissue; lanes 7–9 = infected pistils. Tissues infected with *B. cinerea* consistently showed the double-banding pattern.

**Figure 3 ijms-25-00014-f003:**
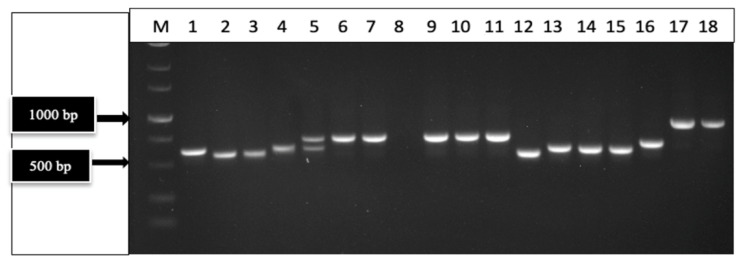
PCR with universal eukaryotic primers for the ITS1-5.8S-ITS2 region of ribosomal DNA shows the amplification of DNA from a range of fungal/oomycete cultures isolated from cannabis plants to produce PCR products of different molecular weight sizes. In addition, cannabis leaf tissues were included for analysis. Lanes 1, 4 = *Trichoderma asperellum*; 2, 3 = *Fusarium oxysporum*; 5 = *Golovinomyces* infected leaf sample; 6, 7, 9–11 = asymptomatic leaf tissues; 8 = blank; 12 = *Botrytis cinerea*; 13–14 = *Fusarium oxysporum*; 15 = *F. proliferatum*; 16 = *Penicillium olsonii*; 17–18 = *Pythium myriotylum*.

**Figure 4 ijms-25-00014-f004:**
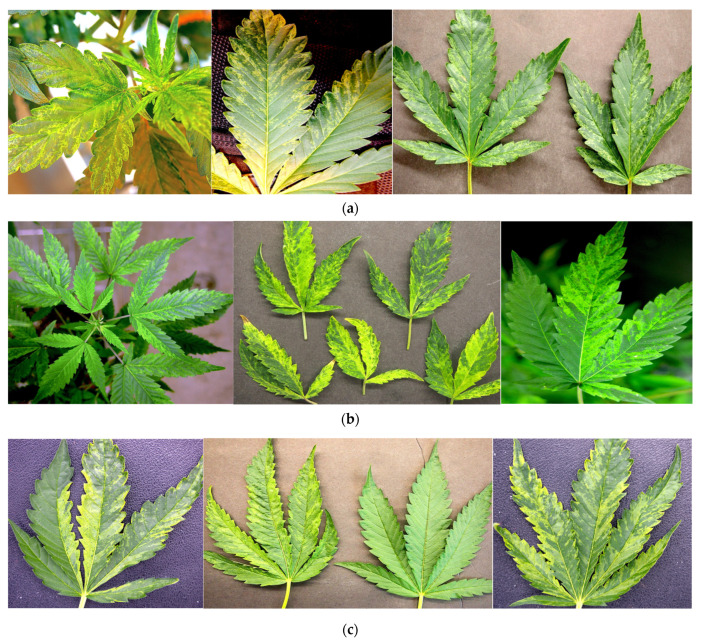
Cannabis leaves displaying symptoms of mosaic and chlorosis putatively attributed to viral/viroid pathogens were subjected to the molecular diagnostic methods described in this study. Four genotypes are shown, namely, ‘OG Kush’ (**a**), ‘Headband’ (**b**), ‘Golden Papaya’ (**c**), and ‘Motor Breath’ (**d**). By comparison, leaves with chlorotic sectors and loss of chlorophyll, characteristic of somatic mutations, are shown in (**e**).

**Figure 5 ijms-25-00014-f005:**
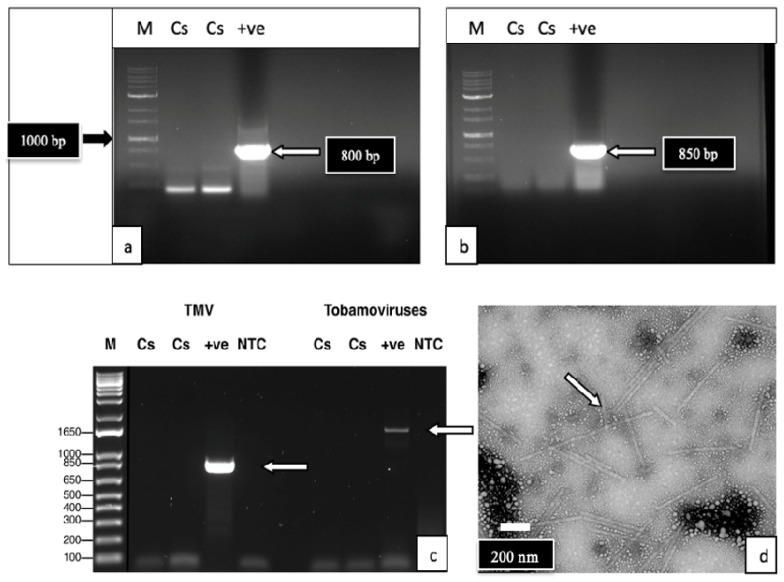
Analysis of cannabis leaf tissues for possible viruses using RT-PCR with specific primers and transmission microscopy. (**a**) Primers for tobacco mosaic virus produced a band at 800 bp (arrow) in the positive control (tobacco) with no corresponding bands in the cannabis leaf samples (Cs). The low-MW bands at around 230 bp were found to be a ubiquitin protein. (**b**) Primers for alfalfa mosaic virus produced a 850 bp band (arrow) in the positive control, but not in the cannabis leaf tissues (Cs). (**c**) Primers for TMV and the Tobamovirus group produced bands at 850 bp and 1650 bp, respectively, in the positive controls (arrows), but not in cannabis leaf samples (Cs). (**d**) Transmission microscopy showed the presence of rod-shaped virus particles measuring 800–1000 nm in the positive control tissue (arrow), but these were not seen in cannabis leaf samples.

**Figure 6 ijms-25-00014-f006:**
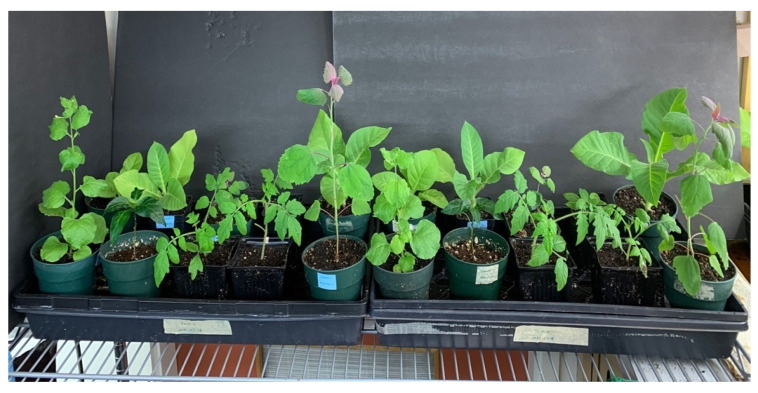
The host range study included *Nicotiana clevelandii* (Cleveland’s tobacco), *N. glutinosa* (Peruvian tobacco), *N. tabacum* ‘Samsun’ (cultivated tobacco), *Chenopodium quinoa* (quinoa), *C. amaranticolor* (goosefoot), *Gomphrena globosa* (globe amaranth), and *Solanum lycopersicoides* (tomato), as shown above. Plants were mechanically inoculated and observed for symptom development after 3 weeks.

**Figure 7 ijms-25-00014-f007:**
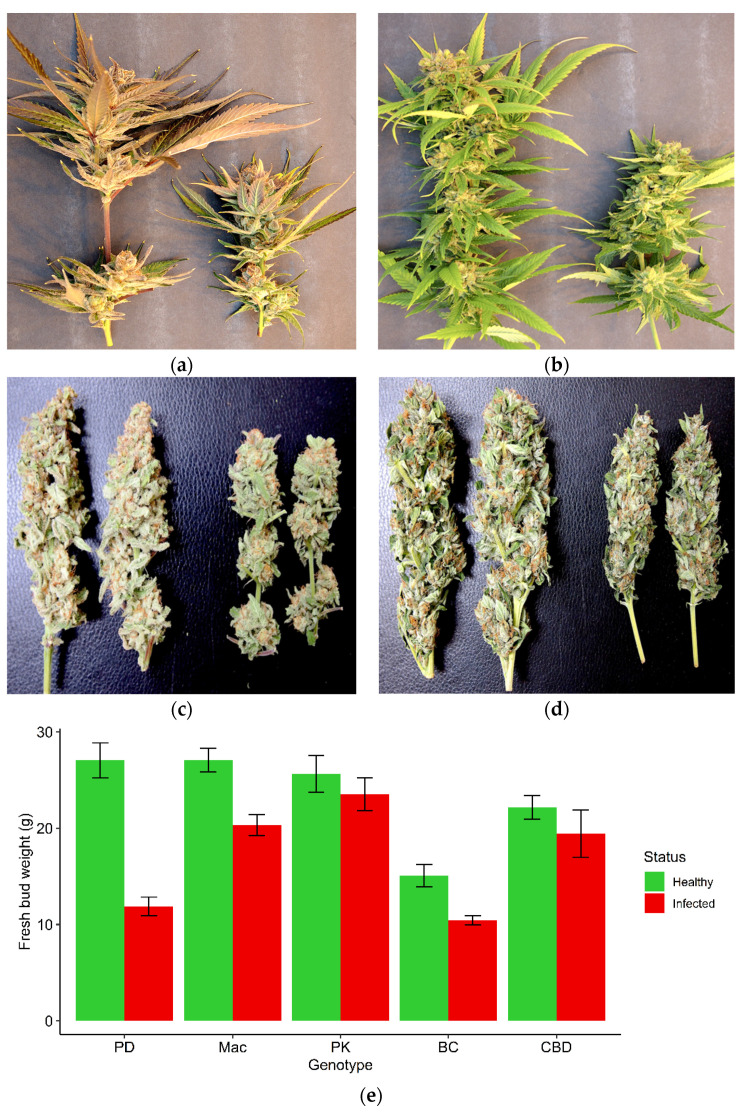
Symptoms of reduced inflorescence growth on two cannabis genotypes attributed to infection by Hop latent viroid, which was confirmed by high-throughput sequencing (HTS). Stunted inflorescence growth can be seen on the genotypes Mac-1 (**a**,**c**) and PD (**b**,**d**). In each photo, the healthy (asymptomatic) samples are shown on the left, while the infected ones are shown on the right. The harvested and trimmed dried inflorescences of the same two genotypes (**c**,**d**) showed a reduced size and volume of the tissues, which was reflected in the reduced weight. (**e**) Fresh weight measurements of the inflorescences of five cannabis genotypes that were healthy (asymptomatic) (green columns) or affected by hop latent viroid as confirmed by HTS (red columns). Measurements were made at harvest. Error bars show standard errors of the mean (*n* = 5). There was no observable reduction in growth on genotypes PK and CBD, while PD, Mac, and BC showed significantly reduced growth.

**Figure 8 ijms-25-00014-f008:**
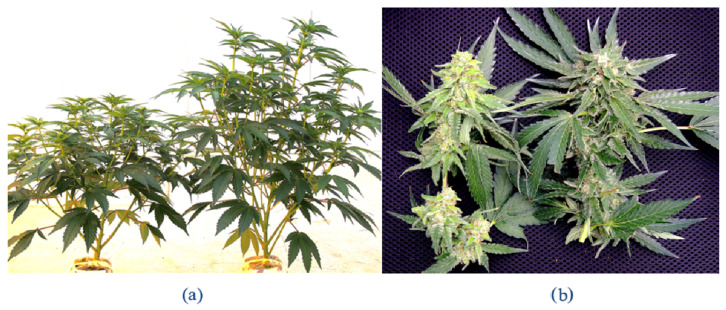
Symptoms of stunted plant growth (**a**), and reduced inflorescence development and chlorosis (**b**), on cannabis genotype PD (left plant in both photos) compared to an asymptomatic plant and healthy inflorescence, respectively (right plant in both photos). The asymptomatic plant was confirmed to be HLVd-negative by RT-PCR.

**Figure 9 ijms-25-00014-f009:**
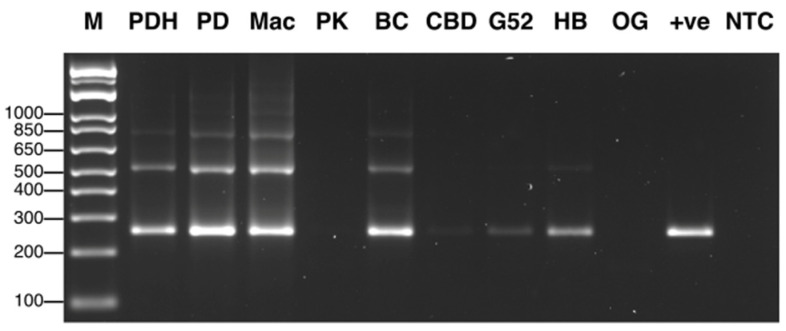
RT-PCR analysis using primers for HLVd shows the viroid was distinctively present in five cannabis genotypes out of nine tested. The lower bands at 256 bp are characteristic for this viroid and represent the entire genome. The larger-sized bands (512 bp) are concatemers of the genome due to its circular nature and represent head–tail alignments. PDH was a healthy (asymptomatic) plant of genotype PD that was subsequently shown to be infected by HLVd, as a band of 256 bp size was present. Genotypes PK and CBD did not show a band, and a very faint band was seen in G54-2. All analyses were conducted on leaf tissues.

**Figure 10 ijms-25-00014-f010:**
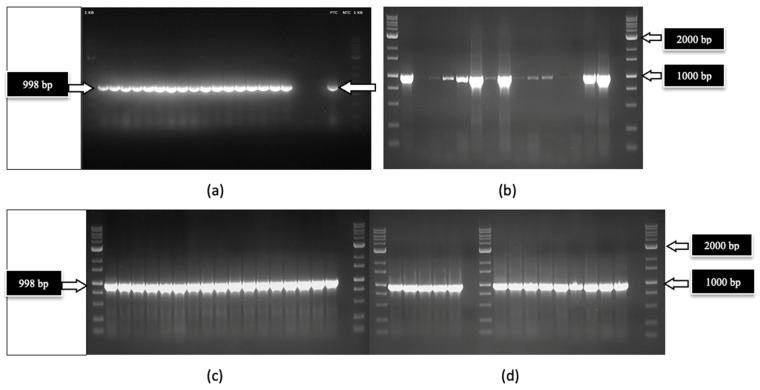
Detection of *Cannabis sativa* mitovirus *1* (CasaMV1) by RT-PCR using specific primers showing the 998 bp band (arrow) was present in leaves of 17 out of 20 genotypes grown indoors (**a**) and in leaves of 10 out of 14 genotypes grown outdoors (**b**). The intensity of the band varied across individual plants in outdoor samples representing four genotypes. In (**c**,**d**), the mitovirus was detected in all samples assayed. These samples originated from four genotypes and consisted of roots, petioles, leaves, and flower tissues from plants grown indoors. The intensity of the bands was consistent across all samples. Blank lanes are water controls.

**Figure 11 ijms-25-00014-f011:**
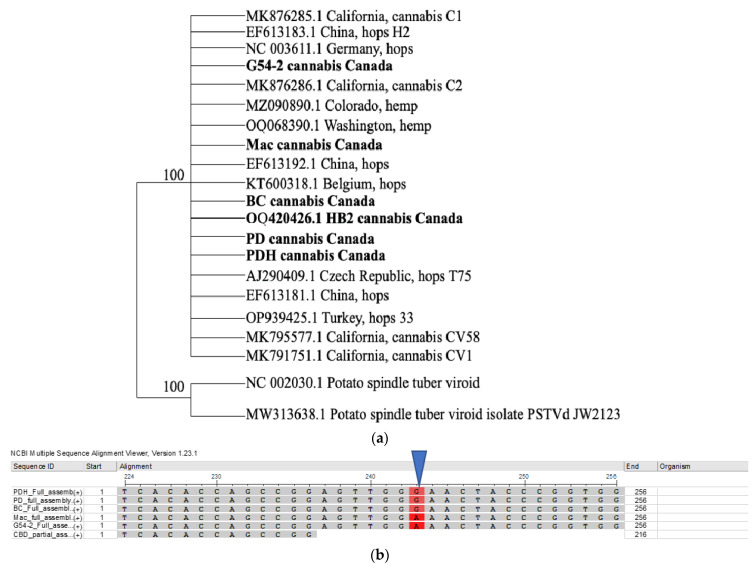
(**a**) Phylogenetic analysis of HLVd strains from cannabis, hemp, and hops including sequences that were obtained from GenBank and from this study (those shown in **bold** were from Canada). The outgroup was PSTVd. (**b**) An SNP was observed in sequences originating from cannabis genotypes ‘Mac’ and G54-2 (arrow).

**Figure 12 ijms-25-00014-f012:**
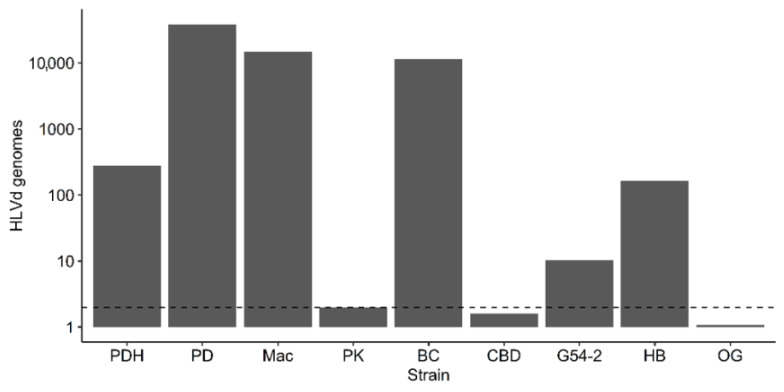
Analysis of HLVd viroid genome copies in the leaf tissues of eight cannabis genotypes as determined by ddPCR. The genotypes PD, Mac, BC, and HB showed obvious symptoms of infection, including stunting and reduced inflorescence development, while genotypes PK, G54-2, CBD, and OG did not display any obvious symptoms. All genotypes were grown adjacent to one another in a commercial greenhouse. Genotype PDH was a leaf sample collected from an asymptomatic, presumed healthy plant growing adjacent to PD in the same row of the greenhouse. The horizontal dashed line is the detection threshold of HLVd—samples measured at or below this threshold are presumed to be HLVd-negative. HLVd measurements were normalized to the cannabis housekeeping gene *EF-1α* and standardized to HLVd levels in the healthy genotype PK.

**Figure 13 ijms-25-00014-f013:**
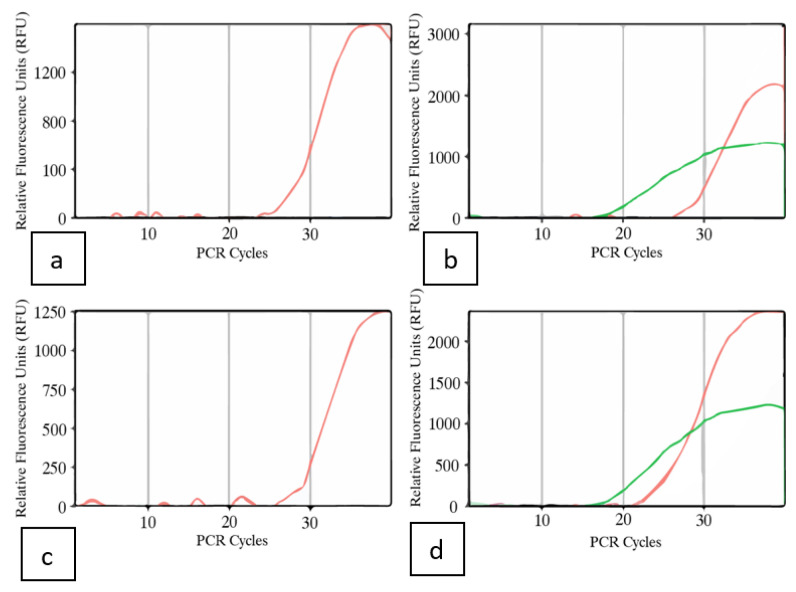
RT-qPCR results showing the relative fluorescence units as a function of PCR cycles (C_T_) in the inflorescence samples of two cannabis genotypes with and without visible symptoms. (**a**) Asymptomatic PD; (**b**) Symptomatic PD; (**c**) Asymptomatic Mac-1; (**d**) Symptomatic Mac-1. Primers used detected the presence of HLVd in both symptomatic samples (**b**,**d**) (green line) compared to the asymptomatic control. Internal control gene cycles are shown with the red lines in each graph. In asymptomatic plants, there was no viroid detected. Positive controls for HLVd were also included for each reaction but are not shown.

**Figure 14 ijms-25-00014-f014:**
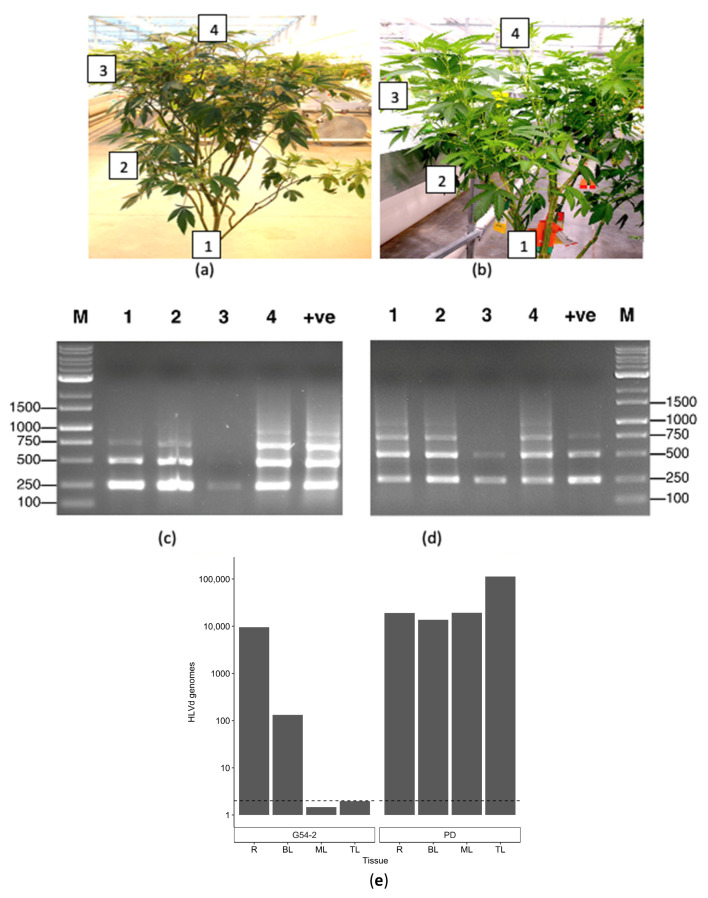
Analysis of 3–4-month-old cannabis stock plants representing two genotypes for the presence of HLVd using RT-PCR and ddPCR. (**a**) Samples of genotype G54-2 were taken from various positions in the canopy (labeled 2–4), as well as from the roots (labeled 1). (**b**) Sampling of genotype PD was conducted in the same manner. (**c**) Genotype G54-2 shows the presence of multiple bands corresponding to HLVd in the samples of the roots (1), bottom-canopy leaves (2), and at the top of the plant (4). In the middle-canopy leaves (3), only a very faint band was observed. (**d**) Genotype PD shows HLVd presence in the samples of the roots (1), bottom-canopy leaves (2), middle-canopy leaves (3), and top-canopy leaves (4). (**e**) The ddPCR of the samples of the roots (R), bottom-canopy leaves (BL), middle-canopy leaves (ML), and top-canopy leaves (TL) of genotypes G54-2 (left) and PD (right). HLVd was detected in the roots and bottom leaves of G54-2, but not in the middle (ML) and top (TL) leaves. HLVd was present in all tissue samples from PD.

**Figure 15 ijms-25-00014-f015:**
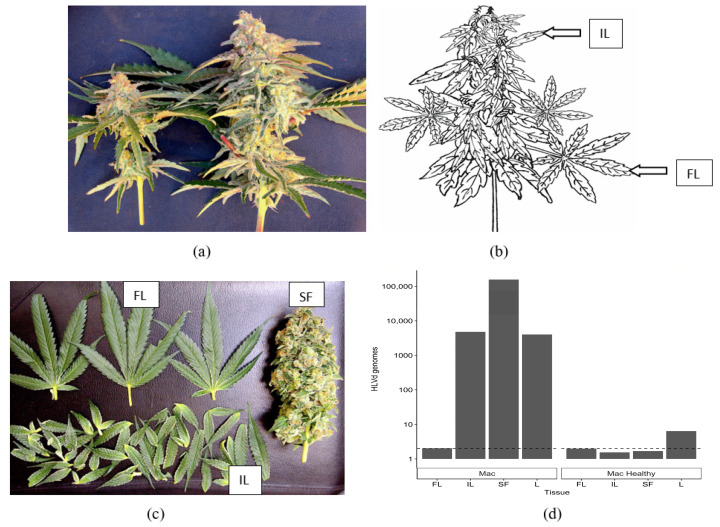
Analysis of the distribution of HLVd in cannabis inflorescence tissues and estimation of viroid genome copies by ddPCR. (**a**) Cannabis inflorescences of genotype ‘Mac-1’ displaying symptoms of infection due to HLVd (left) compared to an asymptomatic (healthy) inflorescence (right). (**b**) The terminal inflorescences were dissected and a schematic drawing shows the distribution of the various tissue types that were sampled. (**c**) Dissection of the fan leaves (FL) and inflorescence leaves (IL) shows the stripped inflorescence flower (SF). All dissected tissues and foliage leaves (PL) were analyzed for HLVd by ddPCR. (**d**) HLVd viroid genome copies seen in different inflorescences tissues shows the highest accumulation in the SF tissues. The horizontal dashed line is the detection threshold of HLVd—samples measured at or below this threshold are presumed to be HLVd-negative. HLVd measurements were normalized to the cannabis housekeeping gene *EF-1α* and standardized to HLVd levels in the healthy genotype PK.

**Figure 16 ijms-25-00014-f016:**
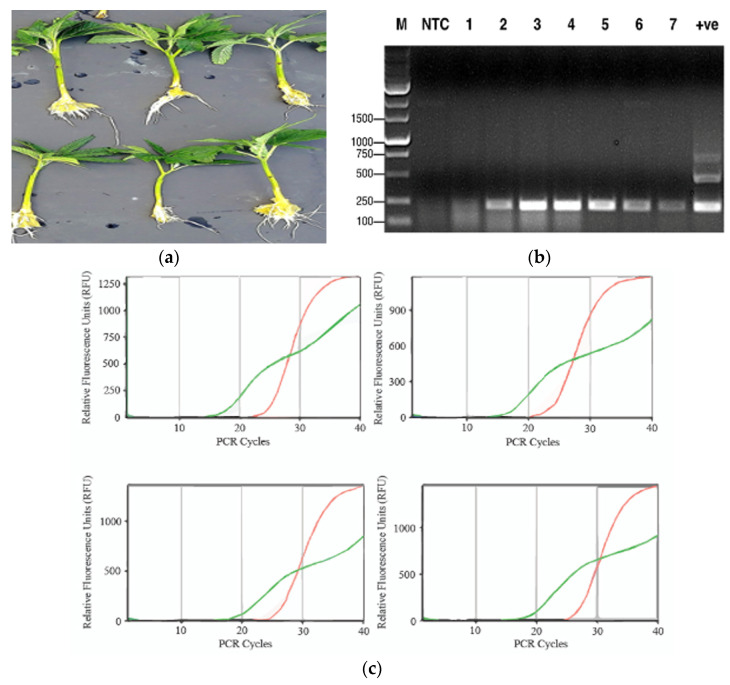
Analysis of emerging roots on cannabis cuttings for HLVd presence by RT-PCR and RT-qPCR. (**a**) The cuttings originated from an infected stock plant of genotype ‘Blue Dream’ and were rooted and sampled after 2 weeks. (**b**) Presence of a 256 bp band characteristic of HLVd was observed in 7 cuttings. Negative control = −ve. Positive control = +ve. (**c**) RT-qPCR with primers used to detect the presence of HLVd in root tissues from 4 infected cuttings (green line). Internal control gene cycles are shown with the red lines in each graph. Positive controls were also included in each reaction. The C_T_ values of 15–18 indicate a high titre of the viroid in the roots of all cuttings.

**Figure 17 ijms-25-00014-f017:**
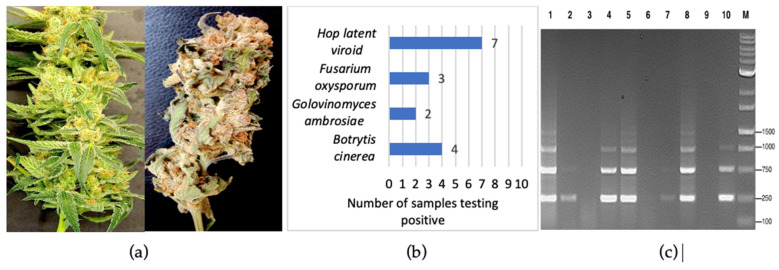
(**a**) Fresh (left) and dried (right) inflorescences from cannabis plants after harvest were subjected to molecular diagnostics in this study. (**b**) From a total of 20 samples, 9 were shown to be infected by fungal pathogens and 7 by HLVd. (**c**) RT-PCR analysis for HLVd presence in 10 dried cannabis samples showed that 7 were positive. Multiple bands on the gels are characteristic for the viroid.

**Figure 18 ijms-25-00014-f018:**
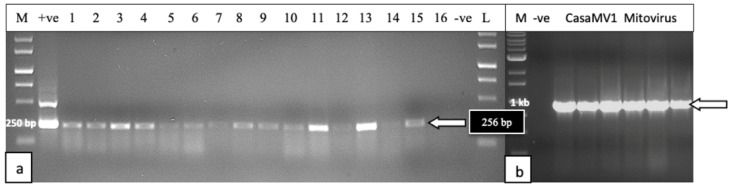
RT-PCR analysis of individual seeds derived from a cross between an infected ‘Mac 1′ female cannabis plant and pollen from a male plant. (**a**) Gel shows presence of HLVd in a majority of the seeds. Whole seeds were used for extraction so it was not determined if the viroid was present on the seed coat or borne internally. There are differences in the band intensity between different seeds, potentially reflecting variable viroid levels between seeds. (**b**) Presence of CasaMV1 was detected in all seeds derived from an infected ‘Mac 1′ female parent that were tested.Seeds were derived from a cross made between an infected ‘Mac1′ female plant and pollen from a male plant of genotype ‘GPie’.

**Figure 19 ijms-25-00014-f019:**
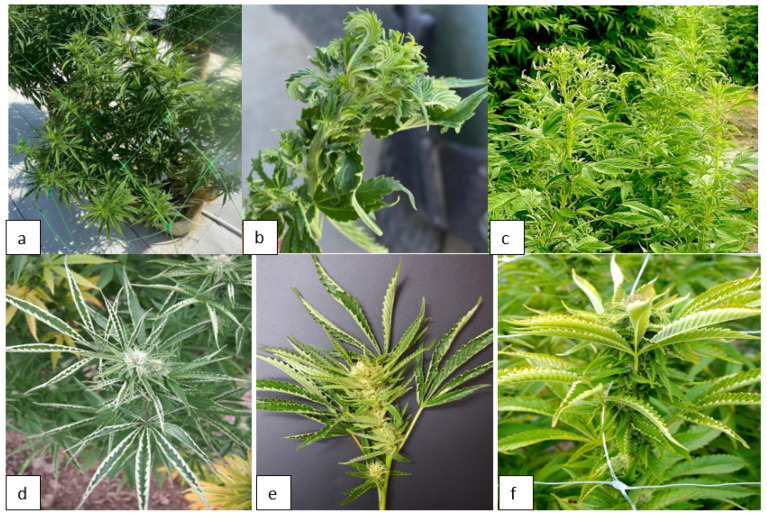
Cannabis plants with symptoms of Beet curly top virus infection were subjected to molecular diagnostics in this study. (**a**,**b**) Indoor-grown plants. (**c**–**f**) Outdoor plants of four different genotypes show varying symptoms attributed to BCTV. Confirmation of virus presence was achieved by RT-PCR with universal primers for BCTV.

**Figure 20 ijms-25-00014-f020:**
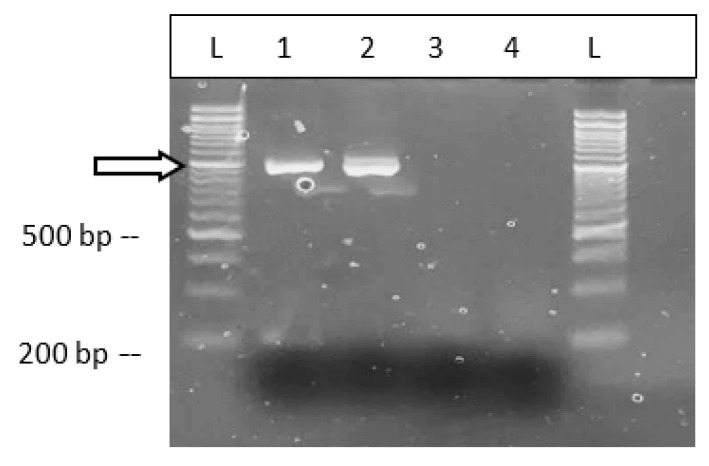
Confirmation of the presence of BCTV in samples of cannabis plants from indoor production. The BCTV-Wor strain was detected (lane 2). Lane 1 = BCTV universal primer; lane 3 = BCTV-Severe; lane 4 = BCTV-Colorado.

**Figure 21 ijms-25-00014-f021:**
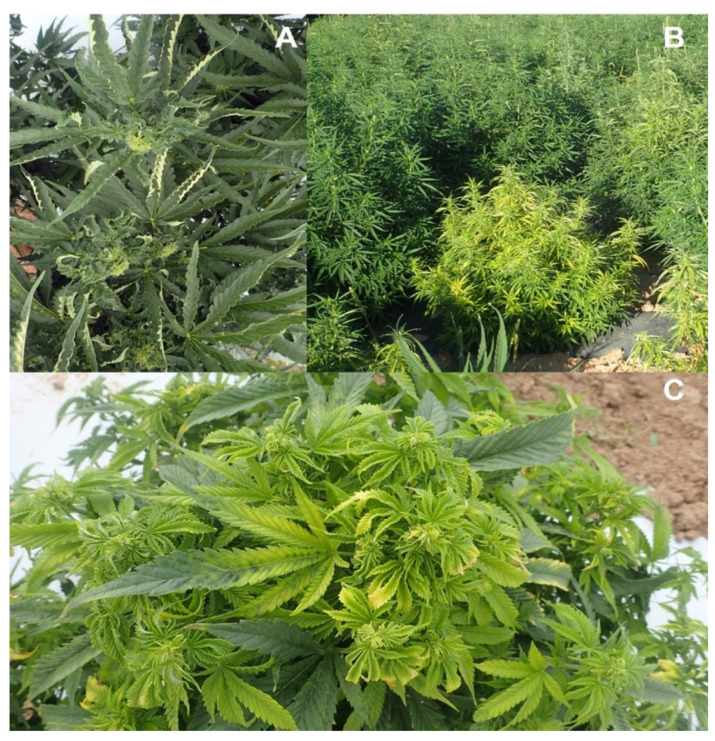
Symptoms of virus/viroid mixed infections on field-grown hemp plants sampled in this study. (**A**) Curling and twisting of upper leaves on infected plants. (**B**) Stunting and yellowing of infected plant. (**C**) Chlorosis, twisting, and mosaic on infected leaves. (Photo credit: Whitney Cranshaw.) The symptoms likely reflect combinations of viruses due to mixed infections in these affected plants.

**Figure 22 ijms-25-00014-f022:**
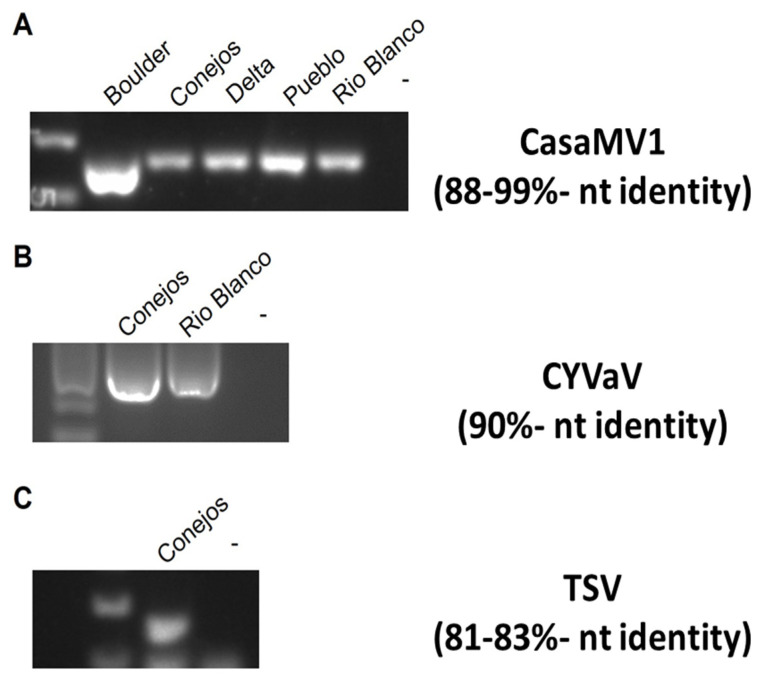
Detection of viruses in hemp plants in Colorado using RT-PCR with virus-specific primers. (**A**) Cannabis sativa mitovirus (CasaMV1). (**B**) Citrus yellow-vein-associated virus (CYVaV). (**C**) Tobacco streak virus (TSV). Water (-) was used as a negative control. Modified from Chiginsky et al. [3].

**Figure 23 ijms-25-00014-f023:**
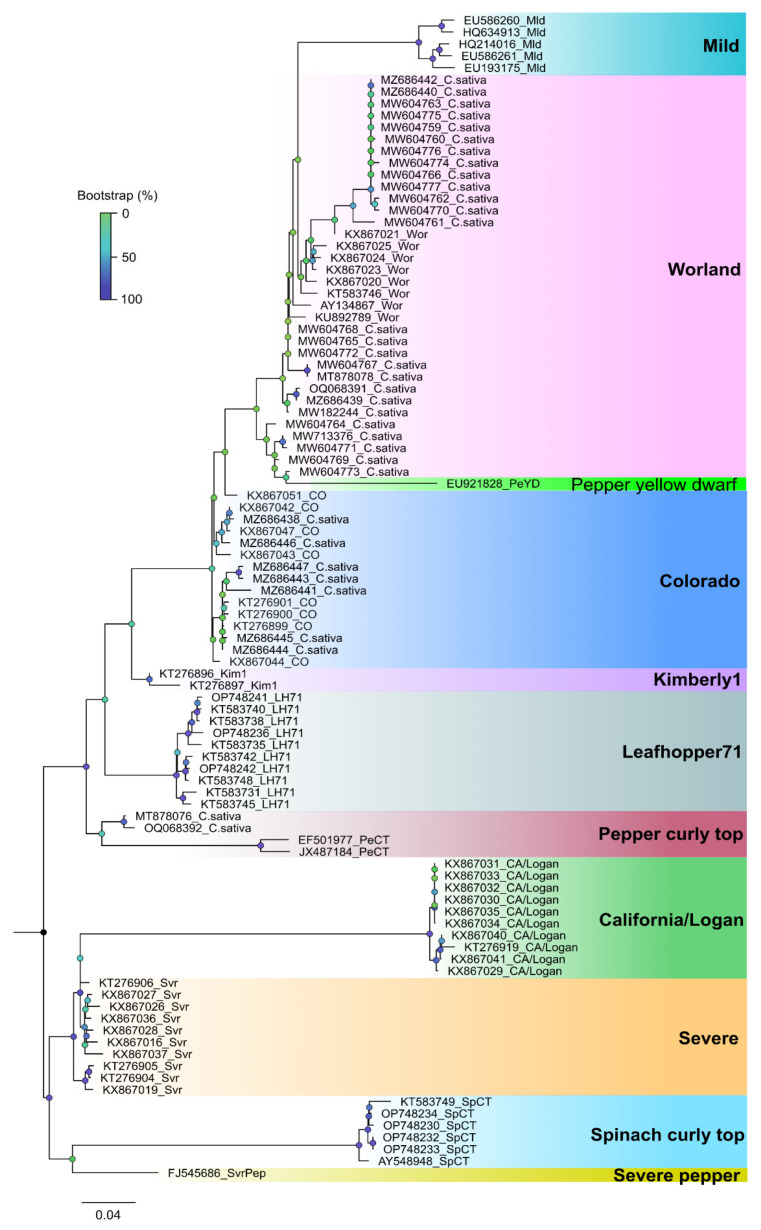
Phylogenetic analysis of partial coat protein sequences of beet curly top virus (BCTV) obtained from hemp samples and other BCTV sequences representing the 11 strains available in GenBank. Multiple sequence alignments were performed using MAFFT v7.505, and the poorly aligned regions were trimmed using TrimAI v1.2.59. The phylogenetic tree was constructed using the maximum likelihood method implemented in the RAxML v8.2.12 with a GTR+G+I model for nucleotide substitution through the CIPRES Science Gateway Environment. BCTV strains, California/Logan; Colorado; Kimberly 1; Mild; Leafhopper 71; Pepper curly top; Pepper yellow dwarf; Severe; Severe pepper; Spinach curly top; Worland.

**Figure 24 ijms-25-00014-f024:**
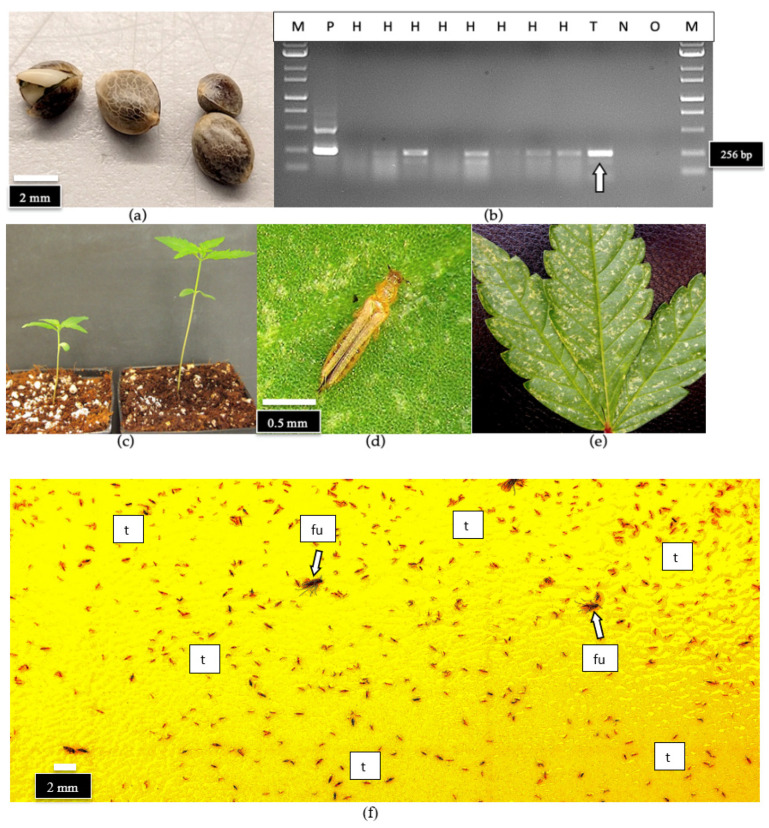
Detection of HLVd on the seeds of hemp and on the thrips feeding on infected hemp seedlings using RT-PCR. (**a**) Seeds were soaked for 24 hr and used in the analysis. (**b**) RT-PCR results of hemp seeds showing that 4 out of 8 seeds tested positive for HLVd and showed the 256 bp size band. Lanes H = hemp seed samples, T = thrips, L = molecular weight ladder, P = positive control for HLVd, N = negative control (no HLVd present), O = water. (**c**) Growth of two seedlings from infected seeds after 2 weeks that both tested positive for HLVd. (**d**) Close-up image of an adult thrip on a hemp leaf (source: Trifecta Natural). (**e**) Extensive thrip damage on a cannabis leaf. Note white (silvery) patches. (**f**) A yellow sticky trap placed in a greenhouse containing flowering cannabis plants shows the high numbers of thrips (t) that may be present. Also shown are fungus gnats (fu, arrow), which have not been implicated in transmission of HLVd but are commonly found in greenhouses.

**Table 1 ijms-25-00014-t001:** Detection of a viroid and virus in cannabis genotypes HB and OG by high-throughput sequencing.

Cannabis	Genotype HB	Genotype HB	Genotype OG	Genotype OG
Pathogen	HLVd	CasaMV1	HLVd	CasaMV1
Genome size (kb)	256	2752	256	2748
Total reads in sample	10,586	402,487	24,120	101,117
Reads per million	4248	161,500	8507	35,664
Reads per million per kb	16,593	58,685	33,231	12,978
Average sequencing depth	4549	19,775	10,241	4912

**Table 2 ijms-25-00014-t002:** Comparative detection of hop latent viroid in leaf, petiole, and root samples of 30 stock plants representing 11 cannabis genotypes using LAMP ^a^.

	Tissue Type Sampled				Tissue Type Sampled		
Cannabis Genotype	Leaf	Petiole	Root	Cannabis Genotype	Leaf	Petiole	Root
PD #1	+	NT	+	BM #1	−	−	−
PD #2	−	NT	+	BM #2	+	+	+
PD #3	+	NT	+	BM #3	−	−	+
PD #4	−	NT	−	BM #4	−	−	+
PK	−	NT	+	BM #5	+	+	+
SC	−	NT	−	BM #6	−	−	+
LM1	+	+	+	111	−	−	−
GP #1	+	+	+	PNW #1	+	+	+
GP #2	−	−	−	PNW #2	−	−	+
GP #3	−	−	+	PNW #3	−	−	+
GP #4	+	−	+	BCP #1	−	NT	+
GP #5	−	NT	+	BCP #2	−	NT	+
GP #6	−	NT	−	BCP #3	+	NT	+
GP #7	−	NT	+	CTQ #1	−	NT	+
G55	−	NT	+	CTQ #2	−	NT	+
Total positive/total	5/15	2/15	11/15		4/15	3/10	13/15

^a^ Leaf samples were selected from the top, middle, or bottom of the plant at random. RNA was extracted from leaf and petiole tissues, as well as from root tissues, of the same plant and subjected to LAMP analysis. (+) indicates positive for HLVd, (−) is negative, NT = not tested.

**Table 3 ijms-25-00014-t003:** Variable presence of HLVd in individual cannabis seeds as determined by Taqman RT-qPCR.

Seed Sample	C_T_ Cycle Threshold	HLVd Present (+)/Absent (−)
Seed #1	−	−
Seed #2	18.94	+
Seed #3	18.56	+
Seed #4	18.36	+
Seed #5	17.10	+
Seed #6	−	−
Seed #7	−	−
Seed #8	28.67	+
Seed #9	33.20	+
Seed #10	33.66	+

**Table 4 ijms-25-00014-t004:** Summary of virus/viroid pathogens identified from hemp fields in Colorado during 2019–2022.

2019	2021	2022
Cannabis sativa mitovirus 1	Cannabis sativa mitovirus 1	Cannabis sativa mitovirus 1
Beet curly top virus, strains CO, BCTV-Wor	Beet curly top virus, strains CO, BCTV-Wor	Beet curly top virus, strains CO, BCTV-Wor
Hop latent viroid	Alfalfa mosaic virus	Tomato bushy stunt virus
Tobacco streak virus	Cannabis cryptic virus	Cannabis cryptic virus
Citrus yellow-vein-associated virus		

**Table 5 ijms-25-00014-t005:** Summary of the various pathogens detected on cannabis and hemp plants in this study using molecular diagnostic approaches.

Molecular Techniques	Pathogen(s) Detected	Detection in Various Tissues
Universal fungal primers	*Fusarium, Pythium, Alternaria, Penicillium, Golovinomyces, Botrytis*	Leaves, stem, root, flower (cannabis)
Virus-group primers	None	
NGS, HLVd-specific primers	Hop latent viroid	Leaves, roots, flower, seeds (cannabis, hemp)
NGS, Mitovirus-specific primers	Mitovirus	Leaves (cannabis, hemp)
NGS, BCTV-specific primers	Beet curly top virus	Leaves (cannabis, hemp)
NGS, CYVaV-specific primers	Citrus yellow-vein-associated virus	Leaves (hemp)
NGS, TSV-specific primers	Tobacco streak virus	Leaves (hemp)
NGS	Alfalfa mosaic virus	Leaves (hemp)
NGS	Tomato bushy stunt virus	Leaves (hemp)

NGS = next-generation sequencing.

**Table 7 ijms-25-00014-t007:** Multiplex Taqman RT-qPCR primers and conditions for the detection of HLVd in this study.

Target	Primer Name	Sequence (5′–3′)	Source
HLVd	HLVd Forward1	ATACAACTCTTGAGCGCCGA	Hataya et al. [75]
	HLVd Reverse1	CCACCGGGTAGTTCCCAACT	Hataya et al. [75]
	HLVd Probe 1	TCTTCGAGCCCTTGCCACCA	This work
	HLVd Forward2	AGTTGCTTCGGCTTCTT	Lu et al. [76]
	HLVd Reverse2	CCATCATACAGGTAAGTCAC	Lu et al. [76]
	HLVd Probe 2	TGCGTGGAACGGCTCCTTCT	This work
Cannabis UBQ	Cannabis UBQ Forward	TACTGCGCCAGCTAACAAAC	Guo [70]
	Cannabis UBQ Reverse	GCACCCGTCTGACCTGAATC	Guo [70]
	Cannabis UBQ Probe	ACAATGCAGCAAATGCTCACTCTACAGCAGTCA	This work

**Table 8 ijms-25-00014-t008:** LAMP primers used in this study for HLVd detection in cannabis tissues.

Primer	Sequence (5′–3′)
HLVd_LAMP_F3	CGAGCTTTACCTGCAGAAGT
HLVd_LAMP_B3	TGAAGAAGGAGCCGTTCCA
HLVd_LAMP_LF	CCCTTGCCACCATACAGG
HLVd_LAMP_LB	CGCGGCGACCTGAAGTT
HLVd_LAMP_FIP	TAGGTTTCCCCGGGGATCCCCCCCTCTGGGGAATACACT
HLVd_LAMP_BIP	CGGAGATCGAGCGCCAGTTCGCAGGACGCGAACAAGAA

**Table 9 ijms-25-00014-t009:** Primer sequences used for detection of Beet curly top virus in cannabis.

Target	Primer	Name	Sequence (5′–3′)	Source
BCTV-Universal	Forward	BCTV2-F	GTGGATCAATTTCCAGACAATTATC	Strausbaugh et al. [77]
	Reverse	BCTV2-R	CCCATAAGAGCCATATCAAACTTC	
BCTV-Worland	Forward	BMCTVv2825	TGATCGAGGCATGGTT	Chen et al. [78]
	Reverse	BGc396	CAACTGGTCGATACTGCTAG	
BCTV-Severe	Forward	BSCTVv2688	GCTGGTACTTCGATGTTG	Chen et al. [78]
	Reverse	BGc396	CAACTGGTCGATACTGCTAG	
BCTV-Colorado	Forward	BCTVCO-F	TGCGAGGACGCTTCTTGATT	Chiginsky et al. [3]
	Reverse	BCTVCO-R	GGGCCGACTCTTATTTTCGG	

**Table 10 ijms-25-00014-t010:** Primers used to identify low-percentage nucleotide-identity viruses in hemp plants in 2019.

Target	Sequence (5′–3′)	Reference
Actin	TTGCTGGTCGTGATCTTACTG GTCTCCATCTCCTGCTCAAAG	Mangeot-Peter et al. [79]
BCTV universal	GCTTGGTCAAGAGAAGT/ CAACTGGTCGATACTGCTAG	Strausbaugh et al. [41]
CasaMV1	GACGTCTTCTTGTTGTGGCTAGTA GTTCATAGGCAACTGAGGTTCTTT	Chiginsky et al. [3]
CYVaV	CCAGACAGGTGTTTCGAGCAT CAATCACTGCAAATCGCG	Kwon et al. [38]
TSV	TGGTGTTGACGAGTAATCGTAGTT GAAGCATTCATCAAACAATAGTCG	Chiginsky et al. [3]

## Data Availability

There were no new data created in this study.

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
