# Peer review of "Challenges to Cannabis sativa Production from Pathogens and Microbes—The Role of Molecular Diagnostics and Bioinformatics"

_ijms, 2023, doi:10.3390/ijms25010014_

Round 1
Reviewer 1 Report
Comments and Suggestions for Authors
This article applies techniques in the area of molecular diagnostics to study pathogens in Cannabis and hemp, mainly including polymerase chain reaction (PCR) - based techniques, such as RT-PCR, multiplex RT-PCR, RT qPCR, and ddPCR, as well as whole genome sequencing (NGS) and bioinformatics. This article exemplifies how these technologies can improve the rapidity and sensitivity of pathogen diagnosis on cannabis and hemp.
The molecular tools used in this paper have promising applications. These tools allow the study of the diversity and origin of specific pathogens, particularly viruses and Viroids. There are some issues in the article that need to be pointed out.
1. The authors did not clearly state the status of the study in the "Introduction" section. They need to refine this part and highlight the main contributions of this paper.
2. The "Discussion" section of this article is too long. Authors are advised to place some basic discussion of results in the "Results" section, which may ease readers' reading pressure.
Author Response
The following revisions have been made in an effort to address Reviewer #1 comments.
- In the Introduction section of the manuscript, just before describing the results, we added in the significant findings from this study and summarize the most important findings.
- The authors have gone through the Discussion section and tried to find areas that can be incorporated into the Results. This leaves a revised Discussion that focuses on just explaining the findings from the work.
Reviewer 2 Report
Comments and Suggestions for Authors
The manuscript "Challenges to Cannabis sativa production from pathogens and microbes – the role of molecular diagnostics and bioinformatics" explores the application of commonly used molecular diagnostic and bioinformatics tools in the detection of pathogens and microbes in cannabis. It emphasizes the positive impact of these molecular biology tools on controlling the quality of cannabis. After a thorough review of the manuscript, I have the following suggestions:
1. I strongly recommend the authors to add scales to figures 5d, 24a, 24c, and 24f. I understand that this may pose a challenge, and if not feasible, I request the authors to provide an explanation.
2. Table 1 lacks borders, making it difficult to read.
3. Many figures do not have DNA ladder annotations indicating molecular sizes.
4. Several bar graphs should have error bars, and I suggest adding them for accuracy.
Author Response
The following changes were made to the manuscript to address the comments from Reviewer #2.
- The requested scale bars were added to each of the requested figures, namely Fig. 5d and Figs. 24 a, c, f. This has made the figures easier to interpret.
- The format of Table 2 was revised and the table was re-done. It is now much easier to read.
- In the figures in which the ladders are unavailable, an insert was added to indicate the exact size of the observed bands in the PCR gels and arrows were added to guide the reader.
- In the bar graphs that the reviewer is referring to and requesting error bars be added, these all represent qualitative data and show, for example, the level of detection of the viroid. As such, error bars cannot be added and no inferences are made as to the statistical differences in the data. They were to merely show patterns of detection levels. Such experiments seldom include replicates due to the qualitative nature of the analysis eg. presence/absence.
Reviewer 3 Report
Comments and Suggestions for Authors
Dear the Editor
Punja ZK et al provided a comprehensive review for the molecular detection of Cannabinoid pathogens. This article seemed suitable for a review rather than a regular research article. This is largely because of its length of text. If this is a simple method paper for diagnosing viral/viroid pathogens for Cannabinoids, please write more concisely based on the instruction for authors. Overall, this manuscript only asked relatively small specific research questions of this area.
Author Response
The submitted manuscript was intended to include original data derived from experimentation in four different laboratories situated in North America to address molecular diagnostics of pathogens affecting cannabis. At the time of preparation of this manuscript, there were merely 3-4 published papers in the literature on this topic. Therefore, a Review article would not have been possible as it would merely be 2-3 pages in length. Instead, the authors chose tp share the methods they have developed, some of which were proprietary and unpublished, to allow the broader research community access to this previously unpublished information. Therefore, there was a high level of detail to be added, which include a description of extensive methodology and results. Admittedly, this has extended the length of the manuscript. The contributions made from the research have addressed several specific research questions regarding the composition of the pathogens affecting cannabis, and more importantly, the molecular diagnostic methods to detect them. Not surprisingly, the significant number of views and downloads of the Preprint version of this paper is a testament to how important this paper will be going forward. It is likely to be a important addition to the literature on cannabis pathology, despite its unavoidable length. The figures in particular are quite illuminating. We hope Reviewer 3 sees the overall merits of this contribution despite the length of the manuscript.
Round 2
Reviewer 3 Report
Comments and Suggestions for Authors
Dear the Editor
This Reviewer understands all data are relevant for this research area, thus these need to be fully exposed.